EMBO
Molecular Medicine

# Histone demethylase KDM4B epigenetically controls NLRP3 expression to enhance inflammatory responses

Li Tong[1,6], Hui Song[1,2,6], Yuan Gao[1], Danhui Qin[1], Caiwei Wang[1], Qi Li[1], Yue Fu[1,2], Chunyuan Zhao[1,2], Zhendong Ying[3], Dailing Chen[3], Chengjiang Gao [ID][1], Chaofeng Han [ID][4], Wei Zhao [ID][1,2,4,5 ✉], Ying Qin [ID][1,2 ✉] & Lei Zhang [ID][3 ✉]

## Abstract

NLRP3 inflammasome, the archetypical molecular driver of inflammation, plays crucial roles in host defense and maintaining cellular homeostasis. Demethylation of histone 3 lysine 9 tri-methylation (H3K9me3, the repressive mark for euchromatic genes) is essential for activating gene transcription. However, whether H3K9 demethylation is required for the induction of proinflammatory cytokines remains largely unknown. Here, we show that histone demethylase lysine-specific demethylase 4B (KDM4B) mediates H3K9me3 demethylation at the *Nlrp3* promoter to induce NLRP3 expression, thereby selectively enhancing NLRP3 inflammasome activation without affecting NF-κB activation. Concordantly, both *Kdm4b* deficiency and the selective KDM4 inhibitor ML324 inhibit NLRP3 inflammasome activation and ameliorate NLRP3-dependent inflammatory diseases in vivo. Furthermore, high glucose level upregulates KDM4B, promoting NLRP3 inflammasome activation and IL-1β secretion, thus aggravating aberrant inflammation during viral infections. Our findings reveal the role of H3K9me3 demethylation in initiating inflammation, identify KDM4B as an epigenetic accelerator of NLRP3, and propose that modulating H3K9me3 could represent a targeted anti-inflammatory strategy.

**Keywords** NLRP3; KDM4B; H3K9me3; Epigenetic Modification; NLRP3 Inflammasome
**Subject Categories** Chromatin, Transcription & Genomics; Immunology; Metabolism

## Introduction

Inflammation is a critical defense response that restores cellular homeostasis following infection and tissue damage. Meanwhile, excessive inflammatory responses lead to the occurrence and development of numerous inflammatory disorders and autoimmune diseases (Chen and Nuñez, 2010; Afonina et al, 2017; Raneros et al, 2021). The initiation of inflammation depends on the activation of pattern recognition receptors (PRRs), which sense pathogen-associated molecular patterns (PAMPs) and endogenous danger-associated molecular patterns (DAMPs) to induce the activation of nuclear factor kappa-B (NF-κB) and the production of inflammatory cytokines, such as interleukin-1 (IL-1) and members of the tumor necrosis factor (TNF) family (Medzhitov, 2008). NOD-, LRR-, and pyrin domain-containing protein 3 (NLRP3) is a key PRR that forms inflammasome, which is the central nodule in inflammatory responses (Fitzgerald and Kagan, 2020; Gong et al, 2020). Classical NLRP3 inflammasome activation requires NF-κB activation during the priming step to upregulate the expression of NLRP3, pro-IL-1β, and pro-IL-18. After activation by diverse stimuli, including ATP, nigericin (Nig), monosodium urate (MSU), cholesterol, and microbial toxins, NLRP3 then recruits the adaptor apoptosis-associated speck-like protein containing a CARD (ASC) and cysteine protease pro-caspase-1 (CASP1) to form NLRP3 inflammasome (Guo et al, 2015; Zhao and Zhao, 2020). NLRP3 inflammasome next cleaves CASP1 into its active forms (p10 and p20), which results in the cleavage of pro-IL-1β, pro-IL-18, and gasdermin D (GSDMD), leading to pyroptosis and the secretion of IL-1β and IL-18. As NF-κB and NLRP3 inflammasome are the two archetypical molecular drivers of the inflammatory responses, it is of utmost importance to regulate their activation to intervene in inflammatory disorders and autoimmune diseases.

Emerging evidence demonstrated that epigenetic changes (such as histone modification) critically control inflammatory responses

[1]Key Laboratory of Infection Immunity and Disease Intervention of Shandong Province, and Key Laboratory for Experimental Teratology of the Chinese Ministry of Education, School of Basic Medical Science, Cheeloo College of Medicine, Shandong University, Jinan, Shandong, China. [2]State Key Laboratory for Innovation and Transformation of Luobing Theory; Key Laboratory of Cardiovascular Remodeling and Function Research of MOE, NHC, CAMS and Shandong Province; Department of Cardiology, Qilu Hospital of Shandong University, Jinan, China. [3]Department of Orthopedic Surgery, the First Affiliated Hospital of Shandong First Medical University & Shandong Provincial Qianfoshan Hospital, Shandong Key Laboratory of Rheumatic Disease and Translational Medicine, Jinan, Shandong, China. [4]Department of Histology and Embryology and Shanghai Key Laboratory of Cell Engineering, Naval Medical University, Shanghai, China. [5]School of Public Health, Cheeloo College of Medicine, Shandong University, Jinan, Shandong, China. [6]These authors contributed equally: Li Tong, Hui Song. ✉E-mail: wzhao@sdu.edu.cn; yingqin@sdu.edu.cn; 1818@sdhospital.com.cn

(Feinberg, 2018; Zhang and Cao, 2019). Histone methylation, one of the major covalent modifications of histones, regulates cell differentiation and development of various diseases, such as cancers (Greer and Shi, 2012). In general, the methylation of histone H3K9 (lysine 9 of histone 3) and H3K27 is considered a repressive marker for euchromatic genes and a landmark modification associated with heterochromatin (Greer and Shi, 2012). H3K27 demethylation, catalysed by Jumonji domain-containing protein (JMJD) 3, and ubiquitously-transcribed TPR protein on the X chromosome (UTX), two lysine-specific demethylase (KDM) 6 subfamily members, are critical determinants of NF-κB-driven proinflammatory genes activation (Kruidenier et al, 2012). However, whether H3K9 demethylation is also required for the induction of proinflammatory cytokines remains largely unknown.

In the present study, we identify the lysine-specific demethylase KDM4B (also known as JMJD2B) as an epigenetic accelerator of inflammation by selectively targeting NLRP3. KDM4B binds to the promoter region of *Nlrp3*, mediates demethylation of H3K9me3, and therefore facilitates *Nlrp3* transcription, ultimately activating NLRP3 inflammasome. Concordantly, both *Kdm4b* deficiency and treatment with the selective KDM4 inhibitor ML324 attenuate NLRP3 inflammasome activation and ameliorate NLRP3-related inflammation, with no effect on NF-κB activation. Moreover, we demonstrate that high glucose upregulates KDM4B to promote NLRP3 inflammasome activation and controls the severity of inflammation during viral infection. Therefore, our results uncover the fundamental role of H3K9 demethylation in initiating inflammation and suggest KDM4B as a key epigenetic accelerator of inflammatory responses.

# Results

## ML324 selectively inhibits *Nlrp3* mRNA transcription and inflammasome activation

To investigate the role of H3K9 demethylation in inflammatory responses, we first examined the proinflammatory cytokine expression in mouse primary peritoneal macrophages (PMs) treated with ML324, a KDM4 demethylase inhibitor (Rai et al, 2012), that suppresses H3K9me3 demethylation. ML324 selectively blocked demethylation of H3K9me3, but not that of H3K27me3, while having no influence on cell viability (Fig. EV1A,B). However, ML324 did not affect the phosphorylation of IκB-α and p65 (Fig. 1A) or the secretion of TNF-α and IL-6 (Fig. 1B), indicating that ML324 had no effect on NF-κB activation. Interestingly, ML324 dose-dependently inhibited the IL-1β secretion, GSDMD and CASP1 cleavage, and pyroptosis triggered by NLRP3 inflammasome activator ATP in LPS-primed PMs, while did not affect TNF-α and IL-6 secretion (Figs. 1C–E and EV1C). Simultaneously, ML324 attenuated NLRP3 protein expression, but not that of ASC or CASP1 (Fig. 1E). Similarly, ML324 inhibited the IL-1β secretion induced by Nig, another NLRP3 inflammasome activator, and LPS transfection, a non-classical NLRP3 inflammasome activator (Figs. 1F and EV1D). ML324 also inhibited NLRP3 inflammasome activation in mouse bone marrow-derived macrophages (BMDMs) (Fig. EV1E). A protein microarray of inflammatory cytokines and chemokines demonstrated that ML324 selectively weakened IL-1β secretion induced by NLRP3 inflammasome activation (Fig. 1G). Furthermore, ML324 had no effect on IL-1β secretion induced by flagellin (an NLR family CARD domain-containing protein 4 [NLRC4]

inflammasome activator) or poly(dA:dT) (an absent in melanoma 2 [AIM2] inflammasome activator) (Fig. 1H). These data indicate that ML324 selectively inhibits NLRP3 inflammasome activation, but not other inflammasomes or NF-κB pathway.

As cellular NLRP3 levels are crucial for the assembly and activation of NLRP3 inflammasome, we then examined the role of ML324 in NLRP3 expression. ML324 inhibited the protein expression of NLRP3, but not that of ASC, CASP1, AIM2, or NLRC4, in LPS-stimulated mouse PMs and human THP-1 cells (Fig. EV1F,G). In addition, ML324 selectively inhibited *Nlrp3* mRNA expression with no effect on the *Il1b*, *Tnfa*, or *Il6* mRNA expression (Figs. 1I and EV1H). We then examined the primary transcript of *Nlrp3* using intronic PCR primers. *Nlrp3* primary transcript induced by LPS significantly reduced in ML324-treated mouse PMs (Fig. 1J). In *Nlrp3*-deficient PMs, ML324 treatment did not affect *Nlrp3* transcription and IL-1β expression (Fig. EV1I,J), suggesting that ML324 inhibited inflammasome activation in an NLRP3-dependent manner. Taken together, these data indicat that ML324 selectively suppresses NLRP3 inflammasome activation via inhibiting *Nlrp3* transcription.

## KDM4B enhances *Nlrp3* transcription and inflammasome activation

ML324 is an inhibitor of KDM4 family, which is comsisting of four histone demethylases (KDM4A, KDM4B, KDM4C, and KDM4D) and two pseudogenes (KDM4E and KDM4F) (Berry and Janknecht, 2013). We next constructed the plasmids encoding the KDM4 family to determine which molecules were crucial for NLRP3 transcriptional expression. KDM4B overexpression enhanced NLRP3 expression, while overexpression of the other KDM4 family members did not (Fig. EV2A,B). To further investigate the function of KDM4A-D, we designed small interfering RNAs (siRNAs) targeting mouse *Kdm4a/b/c/d*, respectively, to suppress their endogenous expression in PMs (Fig. EV2C). *Kdm4b* knockdown significantly inhibited *Nlrp3* expression in PMs, while no significant decrease in *Nlrp3* expression was observed with *Kdm4a/c/d* knockdown (Fig. 2A). KDM4B also enhanced NLRP3 expression triggered by p65 overexpression in HEK293T cells (Fig. EV2D), suggesting that KDM4B promoted TLR-induced NLRP3 expression. Intriguingly, LPS markedly induced KDM4B and NLRP3 expression in mouse PMs (Fig. EV2E,F), suggesting KDM4B plays potential roles in LPS-induced inflammation. To evaluate KDM4B physiological function on *Nlrp3* transcription and inflammasome activation, we then generated *Kdm4b*-deficient (*Kdm4b*⁻/⁻) mice (Fig. EV2G–I). *Kdm4b* deficiency significantly decreased NLRP3 expression in LPS-stimulated mouse PMs (Fig. 2B,C). Consistently, *Kdm4b* deficiency inhibited NLRP3 activation-induced IL-1β secretion, GSDMD and CASP1 cleavage, ASC speck formation, ASC oligomerization, and pyroptosis (Figs. 2D–F and EV2J–M). However, *Kdm4b* deficiency did not affect TNF-α and IL-6 expression, or the phosphorylation and total levels of IκBα and p65 (Fig. 2G,H). *Kdm4b* deficiency also had no effect on IL-1β secretion induced by flagellin or poly(dA:dT) in LPS-primed PMs (Fig. 2I). In addition, ML324 lost the ability to inhibit IL-1β secretion and *Nlrp3* transcription in *Kdm4b*-deficient PMs (Figs. 2J and EV2N). Collectively, these data indicated that ML324 selectively suppressed NLRP3 inflammasome activation and *Nlrp3* transcription via KDM4B. Analogously, *Kdm4b*

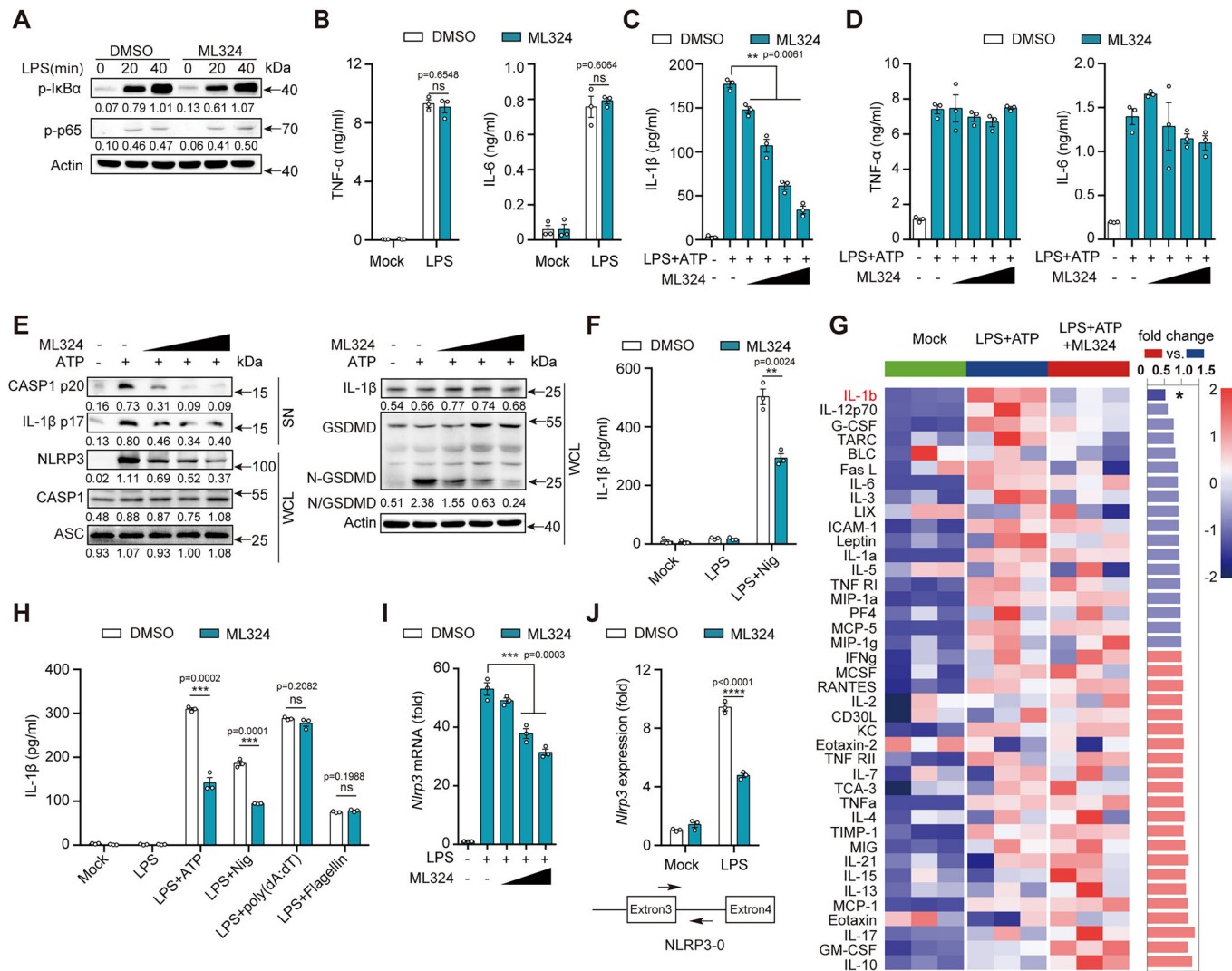

**Figure 1. ML324 inhibits NLRP3 inflammasome activation.**

(A) Western blot analysis of p-IκBα and p-p65 in mouse PMs treated with DMSO or ML324 (20 μM) for 4 h, followed by stimulation with LPS for the indicated time periods. Sizes in kDa are indicated on the right. (B) Enzyme-linked immunosorbent assay (ELISA) analysis of TNF-α and IL-6 secretion in mouse PMs treated with DMSO or ML324 (20 μM) for 4 h, followed by stimulation with LPS for 6 h, n = 3 samples per group. (C, D) ELISA analysis of IL-1β secretion (C), TNF-α and IL-6 secretion (D) in mouse PMs treated with DMSO or increasing amounts of ML324 (5, 10, 20, and 50 μM) for 4 h, followed by priming with LPS for 6 h and the subsequent stimulation with ATP for 40 min, n = 3 samples per group. (E) Western blot analysis of the indicated protein in mouse PMs treated with DMSO or increasing amounts of ML324 (5, 10, and 20 μM) for 4 h, followed by priming with LPS for 6 h and the subsequent stimulation with ATP for 40 min. SN supernatant, WCL whole cell lysate. Sizes in kDa are indicated on the right. (F) ELISA analysis of IL-1β secretion in mouse PMs treated with DMSO or ML324 (20 μM) for 4 h, followed by priming with LPS for 6 h and subsequent stimulation with Nig for 40 min. n = 3 samples per group. (G) Heatmap of protein microarray indicating the inflammatory cytokines and chemokines analysis of mouse PMs treated with DMSO or ML324 (20 μM) for 4 h, followed by stimulation with LPS for 6 h and the subsequent stimulation with ATP for 40 min, n = 3 samples per group. (H) ELISA analysis of IL-1β secretion in mouse PMs treated with DMSO or ML324 (20 μM) for 4 h, followed by priming with LPS for 6 h and subsequently transfected with Flagellin or poly(dA:dT) for 40 min, n = 3 samples per group. (I) Reverse transcription polymerase chain reaction (RT-PCR) analysis of *Nlrp3* mRNA expression in mouse PMs treated with DMSO or increasing amounts of ML324 (5, 10, and 20 μM) for 4 h, followed by stimulation with LPS for 2 h, n = 3 samples per group. (J) RT-PCR analysis of the primary transcripts of *Nlrp3* in mouse PMs treated with DMSO or ML324 (20 μM) for 4 h, followed by stimulation with LPS for 2 h, n = 3 samples per group. Results were obtained from three independent experiments. Data are shown as mean ± SEM in (B–D, F, H–J). ns (not significant), P > 0.05; **P < 0.01; ***P < 0.001; ****P < 0.0001 (two-tailed unpaired *t* test). Source data are available online for this figure.

knockdown reduced NLRP3 expression and selectively inhibited NLRP3 inflammasome activation, while it had no effect on NF-κB activation or TNF-α and IL-6 expression (Figs. 2K,L and EV3A–F). Similarly, *KDM4B* knockdown inhibited NLRP3 inflammasome-induced IL-1β secretion and CASP1 cleavage in human THP-1 cells (Fig. EV3G–I). *Kdm4b* knockdown resulted in the loss of the ability

to suppress *Nlrp3* transcription and IL-1β secretion in *Nlrp3*-deficient PMs (Figs. EV3J,K). Taken together, KDM4B selectively facilitated NLRP3 inflammasome activation in an NLRP3-dependent manner. Furthermore, KDM4B overexpression enhanced *NLRP3* mRNA expression in HEK293T cells, whereas the KDM4B H189A mutant (a demethylase activity-disrupted

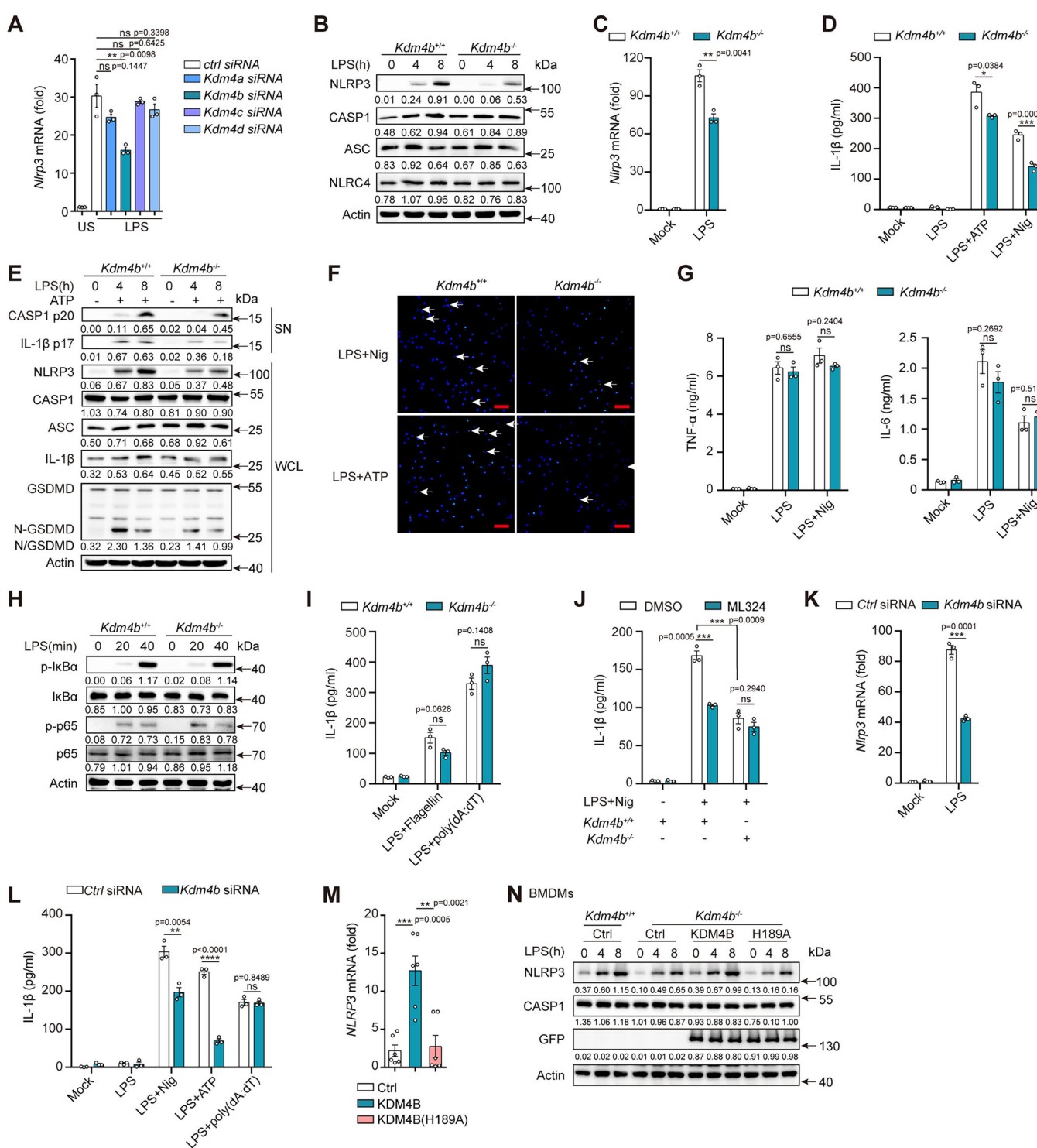

mutant in which a His-to-Ala point mutation was introduced) (Cheng et al, 2018) lost the NLRP3 expression-enhancing effects (Fig. 2M). KDM4B, but not the KDM4B H189A mutant, enhanced NLRP3 inflammasome activation and NLRP3 expression in mouse BMDMs (Figs. 2N and EV3L–N). These data indicate that KDM4B selectively enhanced NLRP3 expression and inflammasome activation via its demethylase activity.

## KDM4B eliminates H3K9me3 at *Nlrp3* promoter region

We next designed six pairs of primers spanning from 2000 upstream to 1500 bp downstream of the NLRP3 transcription start site (TSS) region to determine whether KDM4B binds to the NLRP3 promoter region and the binding region (Fig. 3A). Using chromatin immunoprecipitation (ChIP), we found that KDM4B

◀ **Figure 2. *Kdm4b* deficiency impairs NLRP3 inflammasome activation.**

(A) RT-PCR analysis of *Nlrp3* mRNA expression in Ctrl siRNA- or *Kdm4a/b/c/d* siRNA-transfected mouse PMs, followed by LPS stimulation for 2 h, $n = 3$ samples per group. (B) Western blot analysis of NLRP3, CASP1, ASC, and NLRC4 in PMs from *Kdm4b$^{+/+}$* or *Kdm4b$^{-/-}$* mice after priming with LPS for the indicated time periods. Sizes in kDa are indicated on the right. (C) RT-PCR analysis of *Nlrp3* mRNA expression in *Kdm4b$^{+/+}$* or *Kdm4b$^{-/-}$* mouse PMs stimulated with LPS for 2 h. $n = 3$ samples per group. (D, E) ELISA analysis of IL-1β secretion (D) or western blot analysis of CASP1 cleavage (E) in *Kdm4b$^{+/+}$* or *Kdm4b$^{-/-}$* mouse PMs, followed by priming with LPS for 6 h and subsequent stimulation with ATP or Nig for 40 min. $n = 3$ samples per group. (F) Confocal analysis of the ASC specks in *Kdm4b$^{+/+}$* or *Kdm4b$^{-/-}$* mouse PMs, followed by priming with LPS for 6 h and subsequent stimulation with ATP or Nig for 40 min. The white arrows indicate ASC specks. Scale bars, 50 μm. (G) ELISA analysis of TNF-α and IL-6 secretion in *Kdm4b$^{+/+}$* or *Kdm4b$^{-/-}$* mouse PMs, followed by priming with LPS for 6 h and subsequently stimulated with Nig for 40 min. (H) Western blot analysis of p-IκBα and p-p65 in *Kdm4b$^{+/+}$* or *Kdm4b$^{-/-}$* mouse PMs, followed by stimulation with LPS for the indicated time periods. Sizes in kDa are indicated on the right. (I) ELISA analysis of IL-1β secretion in *Kdm4b$^{+/+}$* or *Kdm4b$^{-/-}$* mouse PMs, followed by priming with LPS for 6 h and subsequently transfected with Flagellin or poly(dA:dT) for 40 min, $n = 3$ samples per group. (J) ELISA analysis of IL-1β secretion in *Kdm4b$^{+/+}$* or *Kdm4b$^{-/-}$* mouse PMs pretreated with DMSO or ML324 (20 μM) for 4 h, followed by priming with LPS for 6 h and subsequent stimulation with Nig for 40 min. (K) RT-PCR analysis of *Nlrp3* mRNA expression in Ctrl siRNA- or *Kdm4b* siRNA-transfected mouse PMs, followed by stimulation with LPS for 2 h, $n = 3$ samples per group. (L) ELISA analysis of IL-1β secretion in Ctrl siRNA- or *Kdm4b* siRNA-transfected mouse PMs, followed by priming with LPS for 6 h and then stimulated with Nig, or ATP or transfected with poly(dA:dT) for 40 min, $n = 3$ samples per group. (M) RT-PCR analysis of *NLRP3* mRNA expression in extracts from HEK293T cells transiently transfected with HA-KDM4B or KDM4B (H189A), $n = 6$ samples per group. (N) Western blot analysis of NLRP3 expression in BMDMs from *Kdm4b$^{+/+}$* or *Kdm4b$^{-/-}$* mice transfected with KDM4B- or KDM4B(H189A)-GFP plasmid, plus stimulation as indicated. Sizes in kDa are indicated on the right. Results were obtained from three independent experiments. Data are shown as mean ± SEM in (A, C, D, G, I–M). ns (not significant), $P > 0.05$; *$P < 0.05$; **$P < 0.01$; ***$P < 0.001$; ****$P < 0.0001$ (two-tailed unpaired $t$ test. Source data are available online for this figure.

most strongly bound to the two upstream regions of the NLRP3 TSS region (NLRP3-3: −12 to −205, and NLRP3-2: −1183 to −1348) in LPS-induced mouse PMs (Fig. 3A), while the NLRP3-2 region is the site at which NF-κB binds to the *Nlrp3* promoter, suggesting that KDM4B may competitively bind to the *Nlrp3* promoter. The NLRP3-3 and NLRP3-2 regions were modified by H3K9me3 and were attenuated after LPS stimulation (Fig. 3B,C). *Kdm4b* deficiency specifically increased H3K9me3 at NLRP3-3 and NLRP3-2 (Fig. 3B,C). Meanwhile, the H3K9me2 and H3K9me1 levels at NLRP3-3 and NLRP3-2 were decreased in *Kdm4b*-deficient mouse PMs (Fig. EV4A–D). However, H3K9me3 modification was not detected in the KDM4B unbound region of NLRP3-6 ( + 1001 to +1239) and remained largely unchanged in *Kdm4b*-deficient PMs (Fig. 3D). Consistent with these results, a dramatic reduction of RNA polymerase II recruitment was observed for NLRP3-3 in *Kdm4b*-deficient and ML324-treated mouse PMs (Fig. 3E,F). In conclusion, KDM4B binds to the *Nlrp3* promoter regions, and demethylates H3K9me3 of the *Nlrp3* promoter regions (NLRP3-3: −12 to −205, and NLRP3-2: −1183 to −1348), thereby enhancing the binding of RNA polymerase II to the *Nlrp3* promoter region, thus promoting *Nlrp3* transcription.

## ML324 and *Kdm4b* deficiency ameliorate inflammation in vivo

Next, we further investigated the physiological roles of KDM4B in inflammation in vivo. IL-1β secretion is induced by LPS intraperitoneal (i.p.) injection, which is NLRP3-dependent (Coll et al, 2015; Qin et al, 2021). The serum levels of IL-1β decreased in ML324-treated LPS-challenged mice, whereas TNF-α and IL-6 remained largely unchanged (Fig. 4A). Alum-induced peritonitis is an NLRP3-dependent inflammation model (Guarda et al, 2011). Mouse peritonitis was induced by i.p. injection of alum. Peritoneal exudate cells (PECs) were collected, and the alum-induced recruitment of inflammatory cells was analyzed by flow cytometry. ML324 administration decreased the number of PECs, neutrophils, and Ly6C$^+$ cells in the abdominal cavity of mice following i.p injection of alum (Fig. 4B). Similarly, ML324 also decreased the number of neutrophils and Ly6C$^+$ cells in the abdominal cavity of mice in the MSU-induced peritonitis model (Fig. 4C). Folic acid

(FA)-induced acute tubular necrosis (ATN) is tightly associated with NLRP3 inflammasome activation (Subramanian et al, 2013). To evaluate the role of ML324 in FA-induced ATN, mice were i.p. injected with ML324, followed by FA. ML324 treatment considerably reduced renal inflammation and edema of kidneys (Fig. 4D–F), and the cleavages of CASP1 and IL-1β in kidneys (Fig. 4G). Concordantly, ML324-treated mice were more resistant to FA injection in survival assays (Fig. 4H). Following LPS challenge, ML324 treatment did not affect the secretion of IL-1β, TNF-α, and IL-6 in serum of MCC950 (an NLRP3 inhibitor) (Coll et al, 2015; Hochheiser et al, 2022)-treated mice, nor did it impact mouse survival rates (Fig. 4I,J). Collectively, these data indicate that ML324 inhibited NLRP3-dependent inflammation in vivo.

We further investigated the physiological relevance of KDM4B in inflammation in *Kdm4b*-deficient mice. *Kdm4b* deficiency markedly suppressed the serum level of IL-1β in LPS-challenged mice, whereas TNF-α, and IL-6 remained unchanged (Fig. 5A). *Kdm4b*-deficient mice were also more resistant to LPS injection in survival assays (Fig. 5B). In the alum-induced peritonitis model, the serum level of IL-1β dramatically decreased in *Kdm4b*-deficient mice, but not the TNF-α and IL-6 levels (Fig. 5C). The alum-induced recruitment of PECs, neutrophils, and Ly6C$^+$ cells was attenuated in *Kdm4b*-deficient mice (Fig. 5D). In addition, lower NLRP3 expression and CASP1 cleavage were observed in the PEC lysates from the *Kdm4b*-deficient group (Fig. 5E,F). ML324 treatment did not significantly affect the secretion of IL-1β, TNF-α, and IL-6 in serum or the survival rate of *Kdm4b*-deficient mice following LPS challenge (Fig. 5G,H). Collectively, these data indicate that *Kdm4b* deficiency ameliorates NLRP3-related inflammation in vivo.

## KDM4B enhances viral infection-induced NLRP3 inflammasome activation

Viral infection induces aberrant NLRP3 inflammasome activation and the cytokine storm, which is closely associated with disease severity, including in coronavirus disease 2019 (COVID-19) (Wang et al, 2014; Rodrigues et al, 2021; Sefik et al, 2022). Viral infection significantly increased KDM4B and NLRP3 expression in mouse PMs (Figs. 6A,B and EV5A). We wonder whether KDM4B-

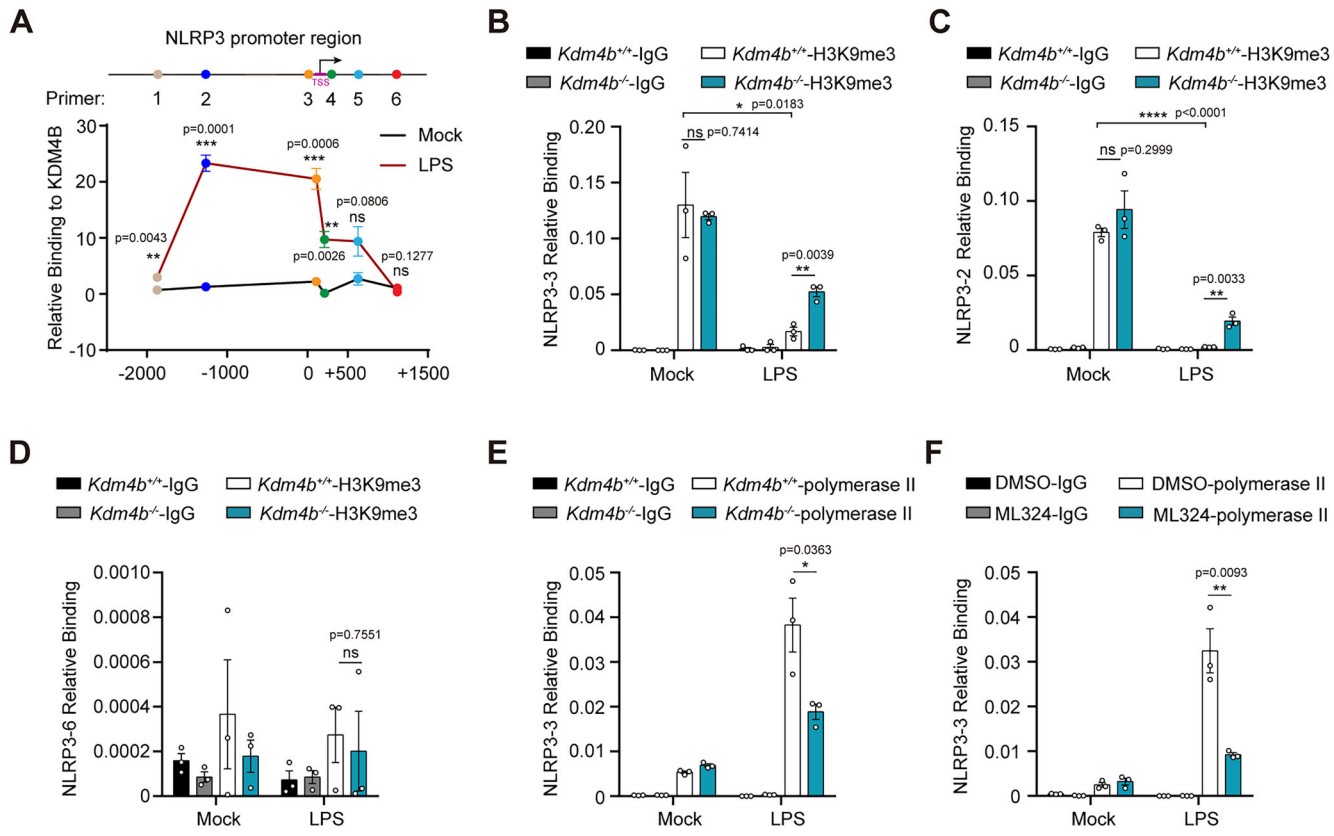

**Figure 3. KDM4B eliminates H3K9me3 at *Nlrp3* promoter region.**

(A) ChIP-qPCR analysis to assess the binding of KDM4B to the *Nlrp3* promoter in mouse PMs stimulated with LPS for 2 h, $n = 3$ samples per group. (B–E) ChIP-qPCR analysis for H3K9me3 (B–D) or pol II (E) at the corresponding region of the *Nlrp3* promoter in mouse PMs from *Kdm4b*[+/+] or *Kdm4b*[−/−] mice stimulated with LPS for 2 h, $n = 3$ samples per group. (F) ChIP-qPCR analysis for pol II at the NLRP3-3 promoter region. Mouse PMs were treated with DMSO or ML324 (20 μM) for 4 h, followed by stimulation with LPS for 2 h, $n = 3$ samples per group. Results were obtained from three independent experiments. Data are shown as mean ± SEM in (A–F). ns (not significant), $P > 0.05$; *$P < 0.05$; **$P < 0.01$; ***$P < 0.001$; ****$P < 0.0001$ (two-tailed unpaired $t$ test). Source data are available online for this figure.

mediated H3K9me3 demethylation affects NLRP3 inflammasome activation during viral infection. KDM4B bound to the *Nlrp3* promoter regions after VSV infection (Fig. 6C). Consequently, *Kdm4b* deficiency suppressed VSV infection-induced IL-1β secretion, CASP1 cleavage, and NLRP3 expression (Fig. 6D–F). While mitochondrial DNA (mtDNA) is recognized as a potent activator of NLRP3 inflammasome (Zhong et al, 2018), *Kdm4b* deficiency exhibited no discernible impact on VSV-triggered mtDNA cytosolic translocation (Fig. EV5B). ML324 increased H3K9me3 at the *Nlrp3* promoter region (Fig. 6G), and reduced the binding of polymerase II to the *Nlrp3* promoter region in VSV-infected macrophages (Fig. 6H). Subsequently, ML324 attenuated NLRP3 expression, the subsequent IL-1β secretion, and CASP1 cleavage caused by VSV infection (Fig. 6I–K).

Next, we examined whether ML324 could ameliorate viral infection-induced inflammation in vivo. Mice were pretreated with ML324 before i.p. injection of VSV (Qin et al, 2023). ML324 treatment reduced the serum level of IL-1β and IL-6, but did not affect TNF-α production (Fig. 6L). Interestingly, the lung tissues from ML324-treated mice exhibited less tissue damage and inflammatory cell infiltration (Fig. 6M). However, ML324-treated and *Kdm4b*[−/−] mice were less susceptible to VSV infection and showed enhanced resistance compared to the WT mice (Fig. 6N,O).

Collectively, these data indicate that *Kdm4b* deficiency inhibits viral infection-induced NLRP3 inflammasome activation, while ML324 could ameliorate viral infection, resulting in less excessive inflammation.

## High glucose upregulates KDM4B to enhance NLRP3 inflammasome activation

Emerging evidence indicates that hyperglycemia is closely related to viral disease susceptibility (Ayres, 2020; Zhang et al, 2022). Elevated glucose levels facilitate the expression of proinflammatory cytokines, including IL-1β, to aggravate uncontrolled inflammatory responses in patients with severe COVID-19 symptoms (Codo et al, 2020). As we have shown that *Kdm4b* deficiency and ML324 relieved viral infection-induced inflammation (Fig. 6), we wondered whether they could improve the susceptibility to inflammation in patients with hyperglycemia by regulating inflammasome activation. We found that high glucose (HG) markedly induced KDM4B and NLRP3 expression (Figs. 7A and EV5C), indicating the potential role of KDM4B in HG-induced aberrant inflammasome activation. HG selectively increased KDM4B binding to the *Nlrp3* promoter region, suppressed H3K9 trimethylation at the *Nlrp3* promoter region and enhanced VSV-induced NLRP3 expression in

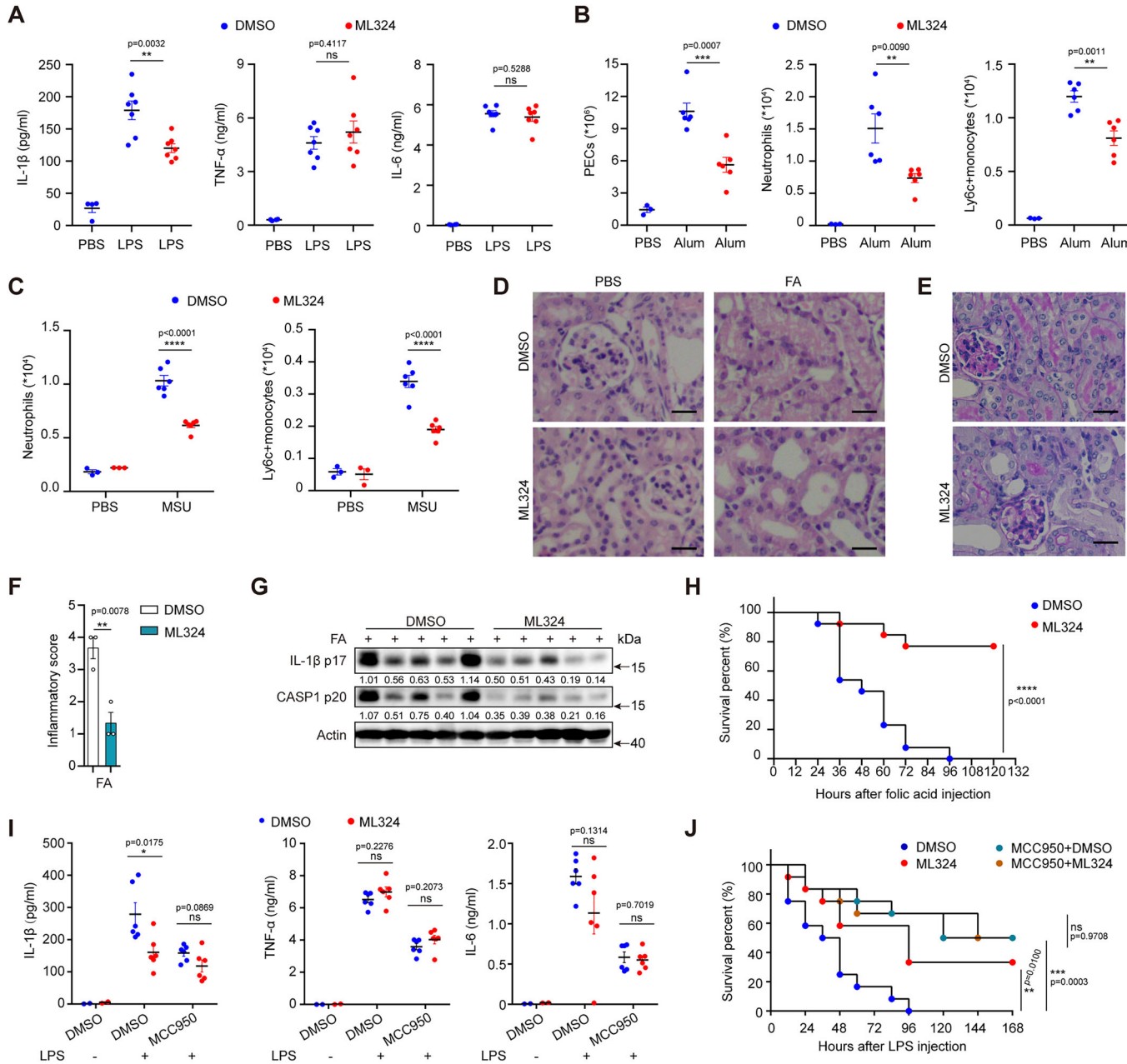

**Figure 4. ML324 ameliorates inflammation in vivo.**

(**A**) ELISA analysis of serum levels of IL-1β, TNF-α and IL-6 from C57BL/6 mice intraperitoneally injected with DMSO or ML324 for 4 h and later with LPS for 90 min. Data were the results of 7 mice in the experimental group ($n = 7$) and 4 mice in the PBS control group ($n = 4$). (**B**) C57BL/6 mice were intraperitoneally injected with DMSO or ML324 for 4 h and later with alum for 16 h. Absolute numbers of peritoneal exudate cells (PECs), neutrophils, and Ly6C$^+$ monocytes recruited to the peritoneum were analysed by flow cytometry. Data were the results of 6 mice in the experimental group ($n = 6$) and 3 mice in the PBS control group ($n = 3$). (**C**) C57BL/6 mice were intraperitoneally injected with DMSO or ML324 for 4 h and then with 1 mg MSU crystals for 6 h. Absolute numbers of neutrophils or Ly6C$^+$ monocytes recruited to the peritoneum were analysed by flow cytometry. Data were the results of 6 mice in the experimental group ($n = 6$) and 3 mice in the PBS control group ($n = 3$). (**D–G**) C57BL/6 mice were initially intraperitoneally injected with DMSO or ML324 for 4 h and later with folic acid (FA) for 36 h. H&E staining of kidney tissue sections (**D**), PAS staining of kidney tissue sections (**E**), renal inflammation was scored based on PAS staining (**F**), Western blot analysis of CASP1 p20 and IL-1β p17 in kidney samples (**G**, $n = 5$). Scale bar 10 μm. Images are representative of four mice per group with similar pathology (**D**, **F**). (**H**) Survival of C57BL/6 mice intraperitoneally injected with DMSO or ML324 ($n = 13$ mice per group) for 4 h then FA. ****$P < 0.0001$ (log rank test [Mantel–Cox]). (**I**, **J**) ELISA analysis of serum levels of IL-1β, TNF-α, and IL-6 from C57BL/6 mice were initially intraperitoneally injected with MCC950 for 24 h, then injected with DMSO or ML324 for 4 h and later with LPS for 90 min (**I**, $n = 6$). Survival of C57BL/6 mice intraperitoneally injected with MCC950 for 24 h, then injected with DMSO or ML324 for 4 h and later with LPS (**J**, $n = 12$). ns (not significant), $P > 0.05$; **$P < 0.01$; ***$P < 0.001$ (log rank test [Mantel–Cox]). Results were obtained from three independent experiments. Data are shown as mean ± SEM in (**A–C**, **F**, **I**). ns (not significant), $P > 0.05$; *$P < 0.05$; **$P < 0.01$; ***$P < 0.001$; ****$P < 0.0001$ (two-tailed unpaired $t$ test). Source data are available online for this figure.

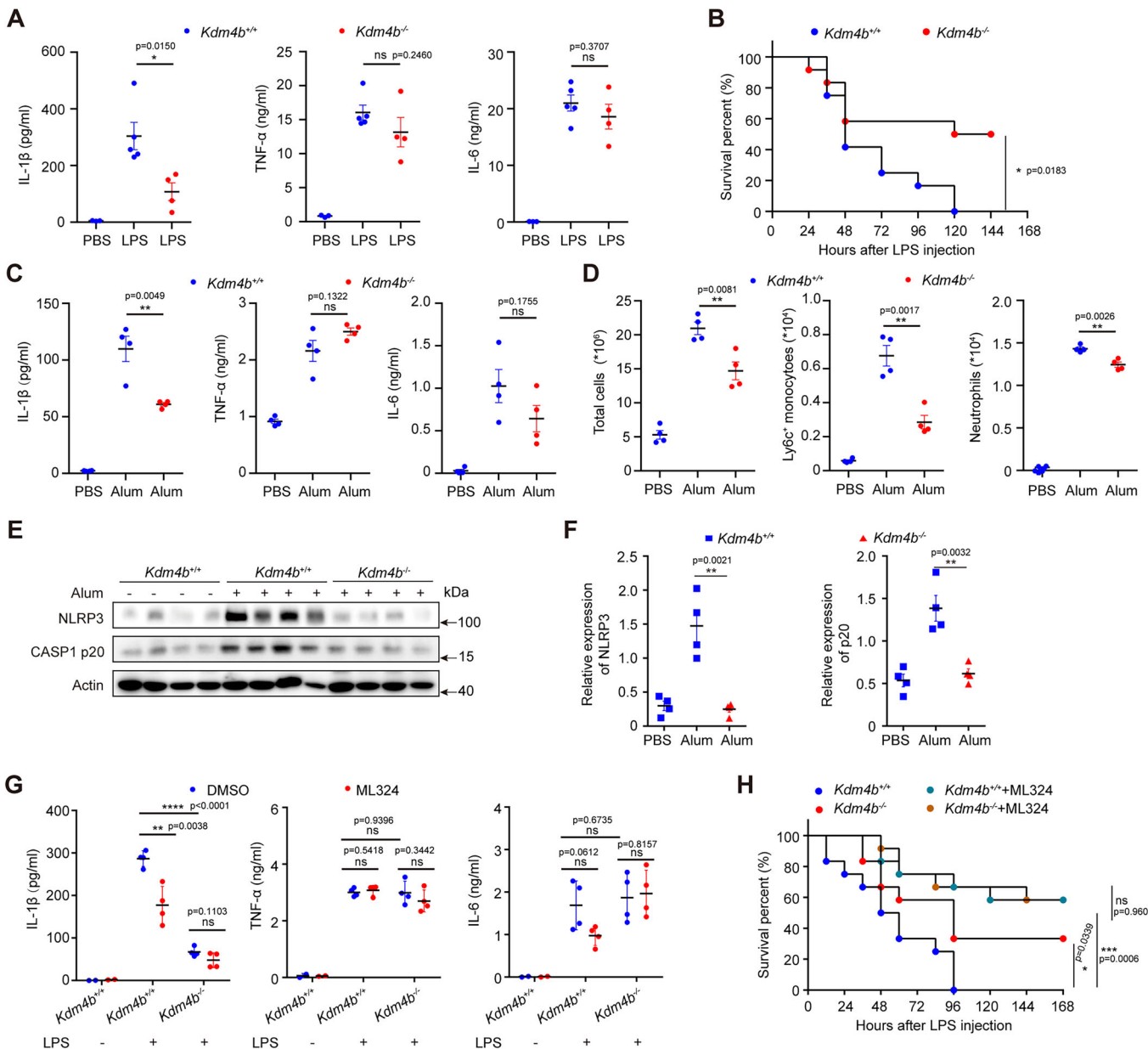

**Figure 5. Kdm4b deficiency ameliorates inflammation in vivo.**

(A) ELISA analysis of serum levels of IL-1β, TNF-α, and IL-6 from Kdm4b+/+ or Kdm4b−/− mice after the intraperitoneal injection with LPS for 90 min. Data were the results of 8 mice in the Kdm4b+/+experimental group (PBS control group n = 3) and 4 mice in the Kdm4b−/− experimental group. (B) Survival of Kdm4b+/+ or Kdm4b−/− mice intraperitoneally injected with LPS (n = 12 mice per group), *P < 0.05 (log rank test [Mantel–Cox]). (C–F) Kdm4b+/+ or Kdm4b−/− mice were intraperitoneally injected with alum for 16 h. ELISA analysis of IL-1β, TNF-α, and IL-6 in the peritoneal lavage fluid (C). Absolute numbers of PECs, neutrophils or Ly6C+ monocytes recruited to the peritoneum were analysed by FCM (D). PECs were lysed and analysed for the expression of NLRP3 and CASP1 p20 using western blot analysis (E). The expression levels of NLRP3 and CASP1 p20 were quantitated by measuring band intensities using the "ImageJ software" (F, n = 4 mice per group). Data were the results of 4 mice in the experimental group (n = 4) and 4 mice in the PBS control group (n = 4). (G, H) ELISA analysis of serum levels of IL-1β, TNF-α, and IL-6 from Kdm4b+/+ or Kdm4b−/− mice were initially intraperitoneally injected with MCC950 for 24 h, then injected with DMSO or ML324 for 4 h and later with LPS for 90 min (G, Data were the results of 4 mice in the experimental group (n = 4) and 2 mice in the PBS control group (n = 2).). Survival of Kdm4b+/+ or Kdm4b−/− mice intraperitoneally injected with DMSO or ML324 for 4 h and later with LPS (H, n = 12). ns (not significant), P > 0.05; *P < 0.05; ***P < 0.001 (log rank test [Mantel–Cox]). Results were obtained from three independent experiments. Data are shown as mean ± SEM in (A, C, D, F, G). ns (not significant), P > 0.05; *P < 0.05; **P < 0.01; ****P < 0.0001 (two-tailed unpaired t test). Source data are available online for this figure.

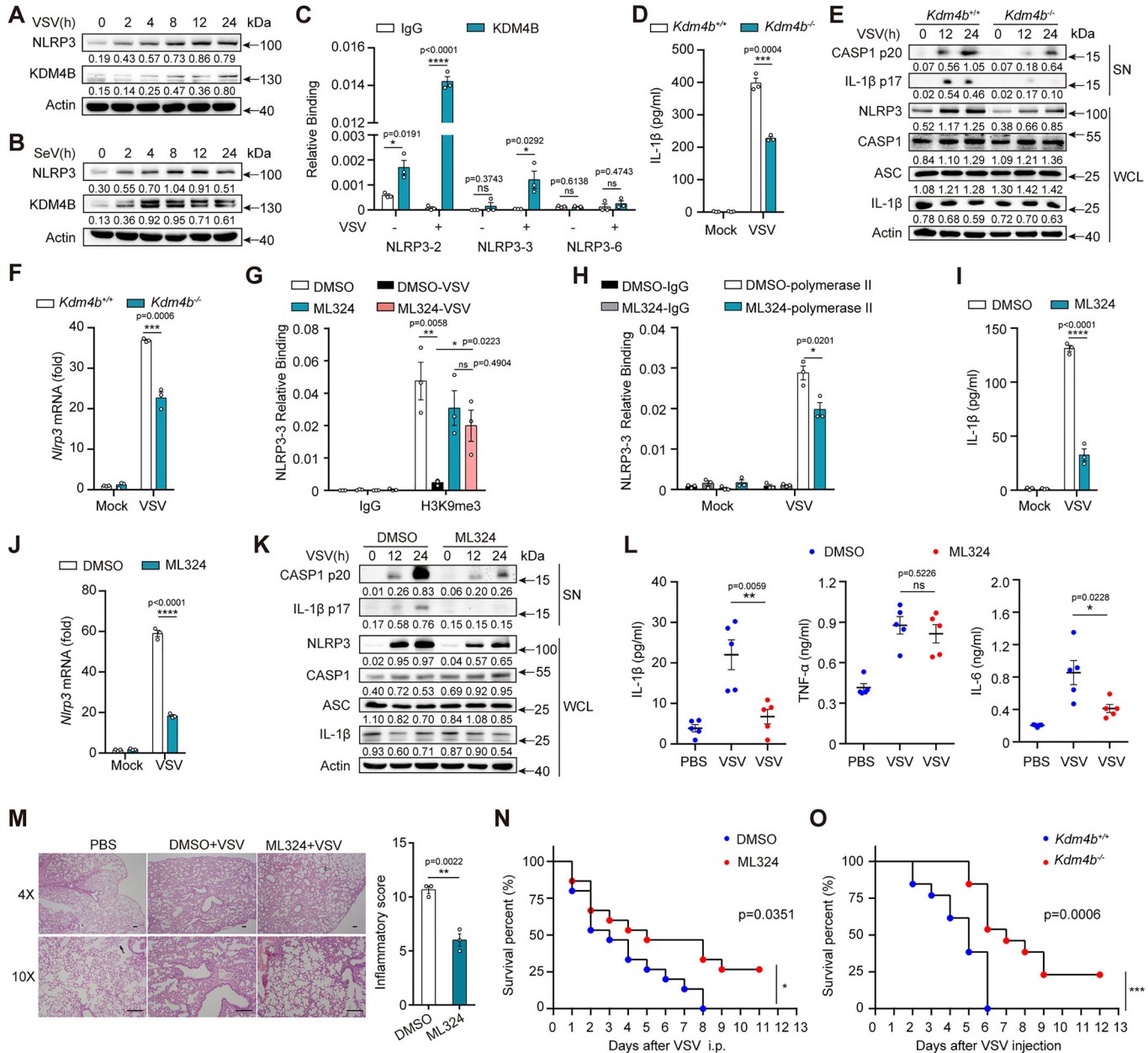

PMs (Fig. 7B–D). Consequently, HG markedly promoted CASP1 cleavage and IL-1β secretion in VSV-infected PMs (Fig. 7E,F). HG also enhanced IL-1β secretion following Nig stimulation in LPS-primed mouse PMs (Fig. EV5D). However, HG did not enhance IL-1β secretion in *Nlrp3*-deficient macrophages (Fig. 7G). *Kdm4b* deficiency under HG had no effect on phosphorylation of IκBα and p65 (Fig. EV5E). Interestingly, both *Kdm4b* deficiency and ML324 treatment reversed HG-induced enhancement of NLRP3 expression and IL-1β secretion in VSV-infected PMs (Fig. 7H–K). Above all, ML324 is a potential therapeutic candidate drug for hyperglycemic patients that may restrain cytokine storm after viral infection. Collectively, these data indicate that KDM4B is crucial for HG-induced NLRP3 inflammasome activation during viral infection (see a model in Fig. 8).

## Discussion

The expression of NLRP3, as a rate-limiting step of canonical activation of NLRP3 inflammasome, is fine-tuned to determine the intensity of inflammation via multiple mechanisms (Swanson et al, 2019). For example, various E3 ubiquitin ligases, such as tripartite motif-containing protein 31 (TRIM31) and membrane-associated RING finger protein 7 (MARCH7), induce Lys(K)48-linked ubiquitination, and the protein degradation of NLRP3, and thus reduce NLRP3 inflammasome-induced IL-1β secretion, while do not affect either TNF-α or IL-6 expression (Song et al, 2016; Yan et al, 2015). MicroRNA (miR)-BART15, induced by miR-223 and Epstein-Barr virus (EBV), and miR-22 specifically target the 3'-UTR (untranslated region) of *Nlrp3* mRNA, inhibiting NLRP3

**Figure 6.  KDM4B enhances viral infection induced NLRP3 expression and inflammasome activation.**

(A, B) Western blot analysis of NLRP3 and KDM4B expression in mouse PMs infected with VSV (A) or SeV (B) for varying periods. Sizes in kDa are indicated on the right. (C) CHIP-qPCR analysis of the KDM4B binding site in the *Nlrp3* promoter infected with VSV, $n = 3$ samples per group. (D) ELISA analysis of IL-1β secretion in *Kdm4b*$^{+/+}$ or *Kdm4b*$^{-/-}$ mouse PMs infected with VSV for 12 h, $n = 3$ samples per group. (E) Western blot analysis of CASP1 cleavage in *Kdm4b*$^{+/+}$ or *Kdm4b*$^{-/-}$ mouse PMs infected with VSV for the indicated time periods. Sizes in kDa are indicated on the right. (F) RT-PCR analysis of *Nlrp3* mRNA expression in *Kdm4b*$^{+/+}$ or *Kdm4b*$^{-/-}$ mouse PMs infected with VSV for 8 h, $n = 3$ samples per group. (G) CHIP-qPCR analysis of the H3K9me3 binding site in the *Nlrp3* promoter in PMs treated with DMSO or ML324 for 4 h, then infected with VSV, $n = 3$ samples per group. (H) *Kdm4b*$^{+/+}$ or *Kdm4b*$^{-/-}$ mouse PMs were stimulated with VSV, and then prepared for ChIP-qPCR assay using pol II Ab, $n = 3$ samples per group. (I) ELISA analysis of IL-1β secretion in mouse PMs treated with DMSO or ML324 (20 μM) for 4 h, followed by VSV infection for 12 h, $n = 3$ samples per group. (J) RT-PCR analysis of *Nlrp3* mRNA in mouse PMs treated with DMSO or ML324 (20 μM) for 4 h, followed by VSV infection for 8 h, $n = 3$ samples per group. (K) Western blot analysis of CASP1 cleavage in mouse PMs treated with DMSO or ML324 (20 μM) for 4 h, followed by VSV infection for the indicated time periods. Sizes in kDa are indicated on the right. (L, M) C57BL/6 mice were intraperitoneally injected with DMSO or ML324 for 4 h and then infected with VSV for 12 h. ELISA analysis of IL-1β, TNF-α, and IL-6 serum levels (L, $n = 5$ mice per group), H&E staining of lung tissue sections (M). Images are representative of 4 mice per group with similar pathology. Lung tissue injury score per field: Each of the 5 indicators was scored individually (0–3 points) in 3 random microscopic fields. The "total score per field" was calculated by summing the scores of the 5 indicators, with a range of 0 to 15 points (M). Scale bar 100 μm. (N) Survival of C57BL/6 mice intraperitoneally injected with DMSO or ML324 ($n = 15$ mice per group) for 4 h and later infected with VSV, *$P = 0.0313$ (log rank test [Mantel–Cox]). (O) Survival of *Kdm4b*$^{+/+}$ or *Kdm4b*$^{-/-}$ mice ($n = 13$ mice per group) intraperitoneally injected with VSV, ***$P < 0.001$ (log rank test [Mantel–Cox]). Results were obtained from three independent experiments. Data are shown as mean ± SEM in (C, D, F–J, L, M). ns (not significant), $P > 0.05$; *$P < 0.05$; **$P < 0.01$; ***$P < 0.001$; ****$P < 0.0001$ (two-tailed unpaired *t* test). Source data are available online for this figure.

expression at the posttranscriptional level and affecting the threshold of NLRP3 activation (Bauernfeind et al, 2012; Haneklaus et al, 2012; Li et al, 2018). NF-κB is a crucial transcription factor that controls NLRP3 transcription. Aryl hydrocarbon receptor binds to the xenobiotic response elements in the NLRP3 promoter region to block NF-κB binding, inhibiting NLRP3 transcription (Huai et al, 2014). The N6-methyladenosine demethylase fat mass and obesity-associated protein (FTO) controls NLRP3 transcription by regulating the phosphorylation of p65 (Luo et al, 2021). H3K9/K14 acetylation and trimethylation of H3K4 (Lecoeur et al, 2020), and H3K27me3 demethylation (Lin et al, 2023) decrease NLRP3 expression to attenuate inflammation. H3K27me3 demethylase JMJD3 modulates NLRP3 transcription by targeting H3K27me3 in the *Nrf2* promoter region (Huang et al, 2020), whereas H3K4 demethylase lysine-specific demethylase 2 (LSD2) inhibits NLRP3 transcription (Kumar et al, 2020). However, the potential role of H3K9me3 demethylation, a repressive epigenetic landmark, in initiating NLRP3 transcription and the subsequent inflammatory responses remains largely unknown. In this study, we showed that the histone demethylase KDM4B-induced demethylation of H3K9me3 was required for NLRP3 transcription and inflammasome activation. While LPS stimulation (Afonina et al, 2017), viral infection (Poeck et al, 2010), and high glucose levels (Guo et al, 2022) induce NF-κB nuclear translocation to drive NLRP3 expression, we demonstrate these stimuli concurrently upregulate KDM4B, which epigenetically enhances NLRP3 transcription through H3K9me3 demethylation at its promoter. Furthermore, ML324, a specific inhibitor of KDM4 family, could also suppress NLRP3 expression and inflammasome activation via KDM4B, thereby identified as a potential therapeutic candidate drug for NLRP3-relevant disorders.

KDM4B catalyzes a variety of histone residues (H3K9me2/3 is the preferred substrate) (Gray et al, 2025; Wilson and Krieg, 2019; Wang et al, 2022). KDM4B is overexpressed in tumors and is required for efficient cancer cell growth (Berry and Janknecht, 2013). KDM4B reduces H3K9me3 marks at the promoters of genes related to endothelial-to-mesenchymal transition (EndMT) to enhance their transcription and regulate EndMT transition (Glaser et al, 2020). High levels of oncometabolites (Sulkowski et al, 2020) suppress KDM4B expression and enhance H3K9me3 levels in

cancer, which masks the presence of specific local H3K9me3 markers and inhibits homology-dependent repair, thereby promoting tumor growth. Loss of KDM4B in mesenchymal stromal cells exacerbates skeletal aging and osteoporosis via increasing H3K9me3 marks at the promoters of skeletal system development-related genes, such as *Runx2* (Deng et al, 2021). Accumulated fumarate inhibits KDM4B activity and increases H3K9 methylation, thus restraining asporin gene transcription, playing vital roles in alveolar bone recovery. Here, we showed that *Kdm4b* deficiency and inactivation both selectively attenuated NLRP3 transcription and ameliorated NLRP3-related inflammatory disorders, including viral infection-induced inflammations. Interestingly, KDM4B had no effect on NF-κB activation or the expression of other proinflammatory cytokines and chemokines, including TNF-α and IL-6. Notably, KDM4B demonstrates cell type-specific regulation of NF-κB signaling, promoting osteoclastogenesis through H3K9me3 demethylation-mediated p65 recruitment at osteoclast gene promoters (Yi et al, 2021), while selectively amplifying basal p65 activity in neural stem cells without affecting LPS-induced transcription (Das et al, 2013). This context-dependent regulatory mechanism warrants further investigation into whether KDM4B-NF-κB crosstalk modulates NLRP3 inflammasome activity in inflammatory pathologies. Meanwhile, LPS stimulation, viral infection, and high glucose levels markedly induced KDM4B expression, suggesting that KDM4B feedback enhances inflammation. Above all, our works reveal the epigenetic roles of histone demethylase KDM4B in regulating NLRP3 inflammasome activation and inflammation. Simultaneously, these findings also shed light on the complexity of NLRP3 inflammasome regulation during viral infections and inflammatory diseases.

In summary, we show that KDM4B epigenetically enhances NLRP3 inflammasome activation by selectively facilitating demethylation of H3K9me3 at the *Nlrp3* promoter region to increase its transcription. High glucose upregulates KDM4B to promote NLRP3 inflammasome activation and IL-1β secretion, thus aggravate viral infection-induced aberrant inflammation. Therefore, we uncover the role of H3K9me3 demethylation in initiating inflammation and provide a novel mechanism by which hyperglycemia causes excessive inflammation and aggravate the severity of viral diseases. Moreover, we demonstrate that *Kdm4b*

deficiency and the selective KDM4 inhibitor, ML324, both ameliorate inflammatory diseases caused by aberrant NLRP3 activity in vivo. As abnormal NLRP3 inflammasome activation is closely associated with multiple inflammatory diseases, our data suggest that ML324 will be a suitable candidate for treating excessive inflammations, and modulating H3K9me3 might be a promising anti-inflammatory strategy with better selectivity.

# Methods

### Reagents and tools table

| Reagent/resource | Source | Catalog number |
|---|---|---|
| **Antibodies** | | |
| Anti-IL-1β/p17 (WB, 1:1000) | Cell Signaling Technology | 12242 |
| Anti-p-IκBα (WB, 1:1000) | Cell Signaling Technology | 2859P |
| Anti-IκBα (WB, 1:1000) | Cell Signaling Technology | 4814S |
| Anti-AIM2 (WB, 1:1000) | Cell Signaling Technology | 13095 |
| Anti-p-p65 (WB, 1:1000) | Cell Signaling Technology | 3031S |
| Anti-p65 (WB, 1:1000) | Cell Signaling Technology | 8242S |
| Anti-Pol II (Chip, 1:200) | Cell Signaling Technology | 4735 |
| Anti-ASC (ASC oligomerization, 1:200) | Cell Signaling Technology | 67824 |
| Anti-Flag (WB, 1:5000, IP, 1:500) | Sigma-Aldrich | F1804 |
| Anti-HA (WB, 1:5000, IP, 1:500) | Sigma-Aldrich | H3663 |
| Anti-Myc (WB, 1:1000) | Origene | 9E10 |
| Anti-NLRP3 (WB, 1:1000) | AdipoGen Life Sciences | AG-20B-0014 |
| Anti-caspase-1 p20 (WB, 1:1000) | AdipoGen Life Sciences | AG-20B-0042 |
| Anti-caspase-1 p10 (WB, 1:1000) | AdipoGen Life Sciences | AG-20B-0044B |
| Anti-ASC (WB, 1:1000) | AdipoGen Life Sciences | AG-25B-0006 |
| Anti- caspase-1 (WB, 1:1000) | GenTex | GTX14367 |
| Anti-β-actin (WB, 1:2000) | Santa Cruz Biotechnology | sc-8432 |
| Anti-IgG (Chip, 1:1000) | Abcam | Ab171870 |
| Anti-IL-1β (WB, 1:1000) | Abcam | Ab234437 |
| Anti-H3K9me3 (Chip, 1:200) | Abcam | Ab8898 |
| Anti-H3K9me2 (Chip, 1:200) | Abcam | Ab1220 |
| Anti-H3K9me1 (Chip, 1:200) | Abcam | Ab9045 |
| Anti-H3K27me3 (Chip, 1:200) | Abcam | Ab6002 |
| Anti-GSDMD (WB, 1:1000) | Abcam | Ab209845 |

| Reagent/resource | Source | Catalog number |
|---|---|---|
| Anti-NLRC4 (WB, 1:1000) | Abcam | Ab201792 |
| Anti-KDM4B (Chip, 1:200) | Bethyl | A301-478A |
| Anti-GFP (WB, 1:2000) | Abmart | M20004 |
| **Flow cytometry antibodies** | | |
| CD11b-FITC (1:200) | BioLegends | 101205 |
| F4/80-PE (1:200) | BioLegends | 123110 |
| Gr-1-PE (1:200) | BioLegends | 108407 |
| Ly6C-APC (1:200) | BioLegends | 128015 |
| **Stimulant/agonist** | | |
| LPS (*Escherichia coli*, serotype O111: B4) 200 ng/mL for mouse PMs, 1 µg/mL for THP-1 cells | Sigma-Aldrich | L4130 |
| ATP (5 mM) | Sigma-Aldrich | A1852 |
| Nig (Nigericin, 50 µM) | Sigma-Aldrich | N7143 |
| Poly (dA:dT) (200 ng/mL) | Invivogen | tlrl-patn |
| Pam3CSK4 (300 ng/mL) | Invivogen | tlrl-pms |
| Flagellin (200 ng/mL) | Invivogen | SRP8029 |
| M-CSF (20 ng/mL) | BioLegend | 576408 |
| Inject Folic acid (FA) | Sigma-Aldrich | F8758 |
| Inject MSU | Sigma-Aldrich | U0881 |
| MCC950 | TargetMol | T3701 |
| Glucose | Sigma-Aldrich | 50-99-7 |
| ChIP grade Protein G Magnetic Beads | Cell Signaling Technology | 9006S |
| Inject Alum | Thermo Fisher Scientific | 77161 |
| Phorbol Myristate Acetate (PMA) | MULTISCIENCES (LIANKE) BIOTECH | CS1001 |
| **Plasmids** | | |
| Flag-KDM4B, HA-KDM4B | OriGene | N/A |
| Flag-KDM4A, Flag-KDM4C, Flag-KDM4D | Vigene Biosciences | N/A |
| Myc-p65 | Vigene Biosciences | N/A |
| GFP-KDM4B | MIAOLING Biology | N/A |
| pSPAX2, pMD2.G | Addgene | N/A |
| **Inhibitor** | | |
| ML324 (0, 5, 10, 20, and 50 µM) | Selleck Chemicals | S7269 |
| **Virus** | | |
| VSV-GFP | X. Cao (Naval Medical University) | N/A |
| SeV | China Center for Type Culture Collection | N/A |
| **Animals** | | |
| *Kdm4b*$^{-/-}$ mice | Cyagen Biosciences Inc. | KOCMP-006 |

| Reagent/resource | Source | Catalog number |
|---|---|---|
| *Nlrp3*$^{-/-}$ mice | The Jackson Laboratory, Bar Harbor. | 021302 |
| Wild-type C57BL/6 mice | Vital River Laboratory Animal Technology Co. | N/A |
| **Cell lines** | | |
| Human THP-1 cells | American Type Culture Collection (ATCC) | N/A |
| HEK293T cells | ATCC | N/A |
| **Chemicals, enzymes, and other reagents** | | |
| DMEM medium (Dulbecco's modified Eagle's medium) | Procell | PM150210 |
| Roswell Park Memorial Institute (RPMI medium) | Procell | PM150110 |
| Glucose-free DMEM | Gibco | 11966025 |
| Lipofectamine 2000 | Invitrogen | 52887 |
| INTERFERin | PolyPlus | 409-10 |
| Polybrene | Solarbio | H8761 |
| DAPI | Servicebio | G1012 |
| Dimethyl Sulfoxide (DMSO) | Sigma-Aldrich | D2650 |
| **Laboratory consumables/Kit** | | |
| Cell Counting Kit-8 (CCK-8) | APExBIO Technology | K1018 |
| KOD-Plus-Mutagenesis Kit | Toyobo | SMK-101 |
| G-Series Mouse Inflammation Array 1 Kit | Ray Biotech, Inc. | GSM-INF-1 |
| RNA fast200 RNA Extraction Kit | Fastagen | 220011 |
| TRIzol Reagent | Invitrogen | 15596018CN |
| TIANamp Genomic DNA Kit | TIANGEN | DP304 |
| BCA protein assay | Pierce | 23228 |
| Mouse IL-1β ELISA Kit | Dakewe Biotech Company Ltd. | 1210123 |
| Mouse TNF-α ELISA Kit | Dakewe Biotech Company Ltd. | 1217203 |
| Mouse IL-6 ELISA Kit | Dakewe Biotech Company Ltd. | 1210602 |
| Human IL-1β ELISA Kit | ABclonal Biotechnology Co. | RK00001 |
| **siRNAs** | | |
| Mouse Kdm4b siRNA | 5'-CGGCCACAUUACCCUCCAA-3' | |
| Mouse Kdm4a siRNA | 5'-GUUGAGGACAGUCUUCCCUUTT-3' | |
| Mouse Kdm4c siRNA | 5'-GCUUGAAUCUCCCAAGAUATT-3' | |
| Mouse Kdm4d siRNA | 5'-UAUCUUGGGAGAUCUAAGCTT-3' | |
| Human KDM4B-1 siRNA | 5'-CAAATACGTGGCCTACATA-3' | |

| Reagent/resource | Source | Catalog number |
|---|---|---|
| Human KDM4B-2 siRNA | 5'-CTCTTCACGCAGTACAATA-3' | |
| Control siRNA | 5'-UUCUCCGAACGUGUCACGU-3' | |

## Mice

*Kdm4b*$^{-/-}$ and *Nlrp3*$^{-/-}$ mice were generated using CRISPR-Cas9-mediated genome engineering on a C57BL/6 background. All the animals were kept under specific pathogen-free (SPF) conditions at 40–70% humidity and under daily cycles of 12 h of light at 23 °C and 12 h of dark at 21 °C. All animals were 6–12 weeks of age, and both male and female mice were used. All animal experiments were performed in accordance with the National Institute of Health Guide for the Care and Use of Laboratory Animals and were approved by the Scientific Investigation Board of the School of Basic Medical Science, Shandong University, Jinan, Shandong Province, China.

## Cells

C57BL/6 mice (male and female, 8–12 weeks old) were injected intraperitoneally (i.p.) with 3% Brewer's thioglycolate broth. Three days later, the peritoneal exudate cells were harvested and incubated. Two hours later, the nonadherent cells were removed, and the adherent monolayer cells were used as peritoneal macrophages (PMs) (Jia et al, 2020). To obtain BMDMs, we used a syringe to blow out the bone marrow of the mouse leg bone, and repeatedly blow it into a single-cell suspension. After removing the red blood cells, it was centrifuged, the supernatant was discarded, and resuspended with Dulbecco's modified Eagle's medium (DMEM) complete culture medium (penicillin 100 U/mL, streptomycin 100 U/mL, M-CSF). The solution was changed on the 3rd and 5th days, and BMDMs were obtained after 7 d of induction. The cells were cultured at 37 °C under 5% $CO_2$ in media supplemented with 10% fetal calf serum (FCS, Invitrogen-Gibco, Waltham, MA, USA). PMs and HEK293T cells were cultured in DMEM. THP-1 cells were cultured at the Roswell Park Memorial Institute (RPMI). The high-glucose medium and its control medium were prepared with glucose-free medium according to normal glucose levels in mammals (Codo et al, 2020).

## Plasmids transfection

All the plasmids' constructs were confirmed by DNA sequencing. Plasmids were transiently transfected into HEK293T cells using the Lipofectamine 2000 reagent (Invitrogen) according to the manufacturer's instructions. Small interfering RNAs (siRNAs) were transfected into mouse PMs using INTERFERin reagent, according to the manufacturer's instructions.

## Lentivirus packaging

The complex components for each transfected 10 cm dish are as follows: Pspax2: 4 μg; PMD2.G: 2 μg; pNH series vector (Ctrl, KDM4B and KDM4B[H189A], purchased from MIAOLING biology): 4 μg. After 48 h of transfection, the viral supernatant was collected. The virus supernatant was then filtered with a 0.45-

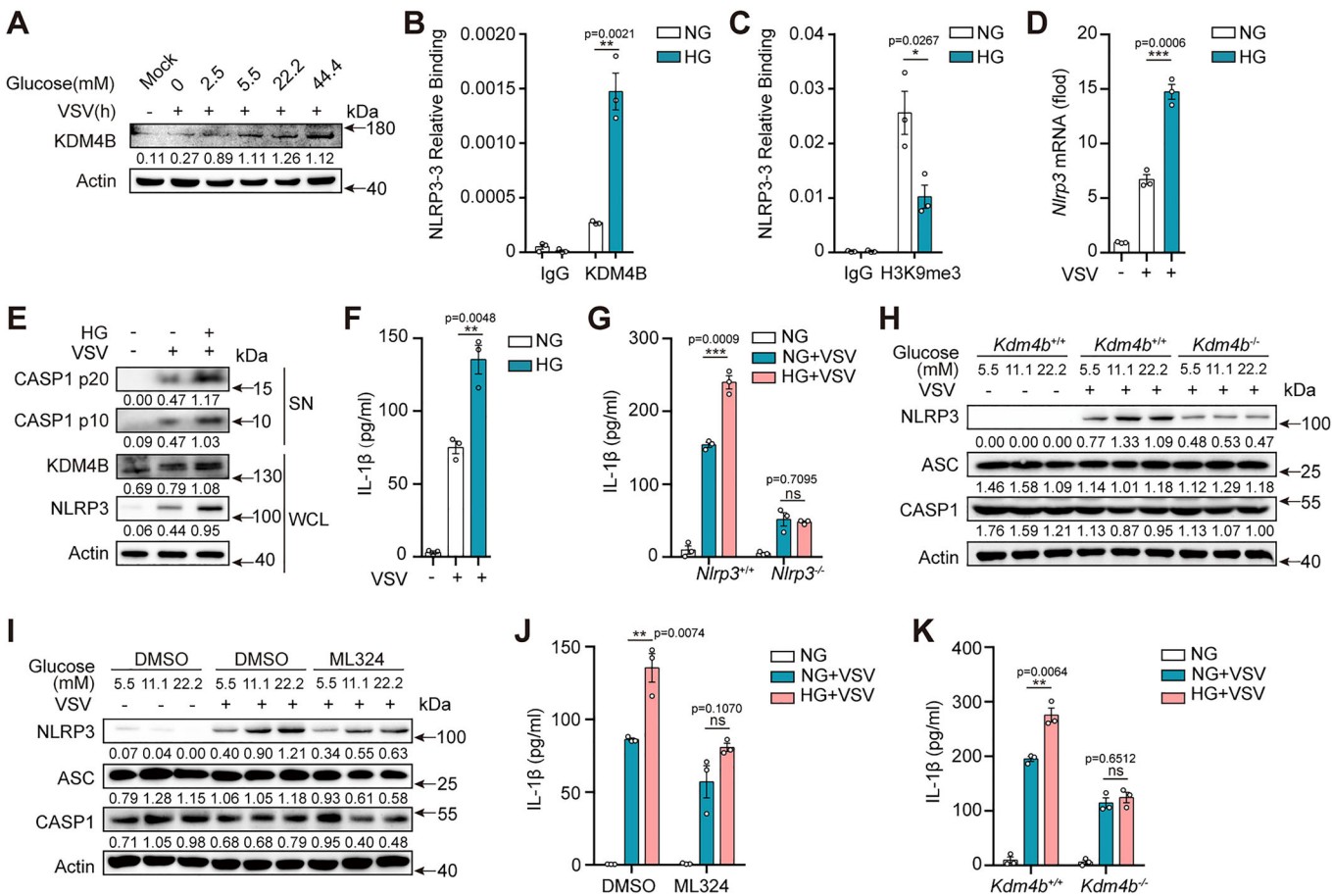

**Figure 7. High glucose upregulates KDM4B to enhance NLRP3 inflammasome activation.**

(A) Western blot analysis of KDM4B in mouse PMs in glucose-free medium treated with increasing concentrations of glucose (0, 2.5, 5.5, 11.1, 22.2, and 44.4 mM) for 12 h, followed by VSV infection. Sizes in kDa are indicated on the right. (B, C) Chip-qPCR analysis of the KDM4B (B) or H3K9me3 (C) binding site in the *Nlrp3* promoter in glucose-free medium treated with normal glucose (NG, 5.5 mM) or high glucose (HG, 11.1 mM). $n = 3$ samples per group. (D–F) Mouse PMs in glucose-free medium were treated with NG or HG for 12 h, followed by VSV infection. RT-PCR analysis of *Nlrp3* mRNA (D, $n = 3$ samples per group), Western blot analysis of CASP1 cleavage (E), ELISA analysis of IL-1β secretion (F, $n = 3$ samples per group). (G) ELISA analysis of IL-1β secretion in *Nlrp3*[+/+] or *Nlrp3*[−/−] mouse PMs in glucose-free medium treated with NG or HG for 12 h, followed by VSV infection. $n = 3$ samples per group. (H) Western blot analysis of NLRP3, ASC, and CASP1 in *Kdm4b*[+/+] or *Kdm4b*[−/−] mouse PMs in glucose-free medium treated with increasing amounts of glucose (5.5, 11.1, and 22.2 mM) for 12 h, followed by VSV infection. Sizes in kDa are indicated on the right. (I) Western blot analysis of NLRP3, ASC, and CASP1 in mouse PMs in glucose-free medium treated with increasing amounts of glucose (5.5, 11.1, and 22.2 mM), then stimulated with DMSO or ML324 (20 μM) for 4 h, followed by VSV infection. Sizes in kDa are indicated on the right. (J) ELISA analysis of IL-1β secretion of mouse PMs in glucose-free medium pretreated with DMSO or ML324 (20 μM) for 4 h, then treated with NG or HG followed by VSV infection. $n = 3$ samples per group. (K) ELISA analysis of IL-1β of *Kdm4b*[+/+] or *Kdm4b*[−/−] mouse PMs in glucose-free medium and then stimulated with NG or HG, followed by VSV infection. $n = 3$ samples per group. Results were obtained from three independent experiments. Data are shown as mean ± SEM in (B–D, F, G, J, K). ns (not significant), $P > 0.05$; **$P < 0.01$; ***$P < 0.001$ (two-tailed unpaired $t$ test). Source data are available online for this figure.

μm filter, centrifuged at 4 °C at 3500 rpm for 35 min. BMDMs are supplemented with 4–8 μg/mL of polybrene treated simultaneously with Lentivirus complex. Polybrene is a positively charged small molecule that binds to anions on the cell surface and improves the efficiency of lentiviral infection of cells, usually the addition of polybrene can increase the infection efficiency by 2–10 times. Lentivirus particles for infection were produced by cotransfecting HEK293T cells with one of the expression plasmids (Ctrl, KDM4B or KDM4B[H189A]) and two packaging plasmids (pMD2G and pSPAX2) at a ratio of 2:1:2 (Wei et al, 2020). Infectious lentivirus particle-containing medium was collected at 48 h after transfection

and centrifuged and filtered through a 0.45-μm filter (Millipore, Burlington, MA, USA).

## ChIP-qPCR

Mouse PMs ($1 \times 10^6$ cells) were cross-linked and cultured at 37 °C for 10 min in 1% formaldehyde, quenched with glycine (5 min rotation at 125 mM), and washed twice with PBS. The cells were lysed using 100 μL SDS lysis buffer (50 mL 0.5 M Tris HCl, 10 mL 0.5 M EDTA, and 5 g SDS, pH 8.0), supplemented with a protease inhibitor mixture and PMSF, and then sonicated (Picoruptor,

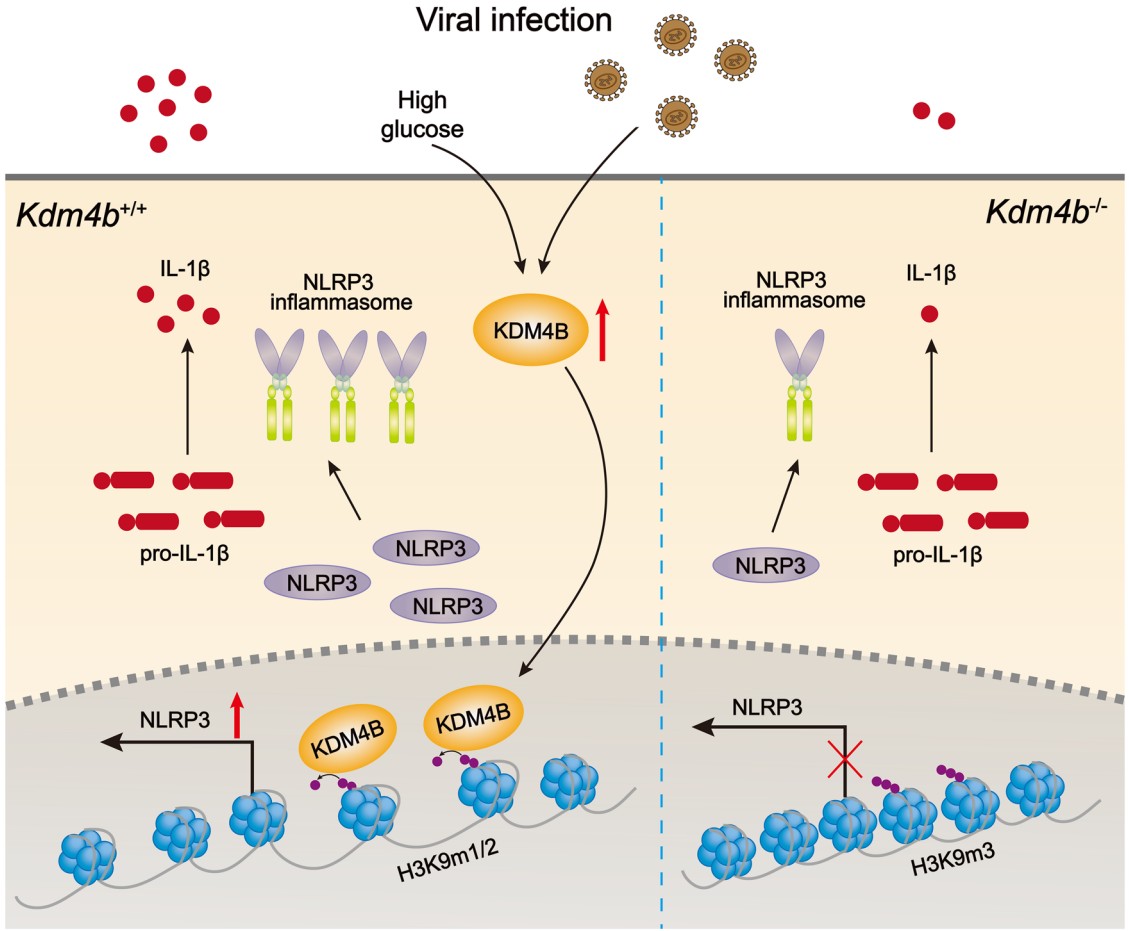

**Figure 8. Schematic representation of the regulatory roles of KDM4B in NLRP3 inflammasome activation.**

High glucose upregulates KDM4B, which binds to the promoter region of *Nlrp3* and mediates the demethylation of H3K9me3, thereby facilitating *Nlrp3* transcription, promoting NLRP3 inflammasome activation, and controlling the severity of inflammation during viral infection. *Kdm4b* deficiency attenuates *Nlrp3* transcription and NLRP3 inflammasome activation and ameliorates NLRP3-related inflammation.

Covaris; settings: 15 s on and 30 s off, 14 cycles). Approximately 100 µL of the supernatant was obtained by centrifugation (12,000 rpm for 10 min), and diluted with 1.8 ml of buffer (0.05 g SDS, 5.5 ml Triton X-100, 1.2 mL 0.5 M EDTA, 16.7 mL 0.5 M Tris HCl, and 4.885 g NaCl [pH 8.0]), then incubated overnight with 2 µL Abs. Following this, 20 µL of protein G from Cell Signaling Technology (Beverly, MA) was added, and the sample was incubated for a further 2 h. The immune complexes were sequentially washed in low salt buffer (4.3875 g NaCl, 5 mL Triton X-100, 2 mL 0.5 M EDTA, and 20 ml 0.5 M Tris HCl [pH 8.0]), high salt buffer (14.625 g NaCl, 5 mL Triton X-100, 2 mL 0.5 M EDTA, and 20 mL 0.5 M Tris HCl [pH 8.0]), LiCl buffer (20 ml 0.5 M Tris HCl, 2 mL 0.5 M EDTA, 2.5 g sodium deoxycholate, 250 mM LiCl, and 0.1% NP-40), and TE buffer (10 mL 0.5 M Tris HCl and 1 mL 0.5 M EDTA [pH 8.0]) twice. The immune complex was acquired using an elution buffer and then reverse cross-linked for 4 h in the presence of 0.2 M NaCl, and 40 µg/ml proteinase K. DNA was obtained and analyzed using RT-PCR with primers specific for *Nlrp3* (1–6) promoter. The fragment sizes of *Kdm4b*[+/+] and *Kdm4b*[−/−] samples were similar, and were analyzed in parallel (mean fragment sizes of ~200–500 bp). All binding results from the IP were normalized according to the input. The antibodies used in the ChIP-qPCR assay were anti-KDM4B, anti-H3K9me3, anti-H3K9me2, anti-H3K9me1, anti-phospho-Rpb1 CTD (Ser2/5), and rabbit IgG, which are all ChIP-grade Abs. The primers used for ChIP-qPCR are listed in Table EV1.

**Protein microarray of inflammatory cytokines and chemokines**

Protein supernatant (150 µL) was used for cytokine and chemokine detection and quantification using a G-Series Mouse Inflammation Array 1 Kit. The array was designed according to the manufacturer's instructions to quantitatively and simultaneously detect 40 inflammatory cytokines, chemokines, and growth factors. The signals (Cy3 channel) were captured using an InnoScan 300

Microarray Scanner (INNOPYS, Carbonne, France). Quantitative data analysis was performed using the RayBiotech mouse Inflammation Array 1 software (GSM-INF-1-SW Analyzer).

## ELISA

The concentrations of mouse IL-1β, TNF-α, IL-6, and human IL-1β were measured using ELISA kits, according to the manufacturer's instructions.

## RT-PCR and western blot

RNA from mouse PMs was extracted using an RNA fast200 RNA Extraction Kit. Total RNA was extracted using TRIzol reagent according to the manufacturer's instructions. The primer sequences used for RT-PCR are listed in Table EV2. Data were normalized to m*Actb* or h*ACTIN* expression in each sample (Zhao et al, 2023). Cells were lysed using radioimmunoprecipitation assay buffer (Pierce, Thermo Fisher Scientific Inc.) supplemented with a protease inhibitor cocktail (Sigma). Protein concentrations were measured using a Pierce BCA Protein Assay Kit. Equal amounts of the extracts were separated by SDS-PAGE and transferred onto PVDF membranes (Millipore, Burlington, MA) for immunoblot analysis. Proteins expression level/Actin was quantitated by measuring band intensities using the "ImageJ" software.

## Isolation of cytosolic fractions and droplet digital PCR (ddPCR)

Cytosolic fractions were obtained by extracting $10^5$ cells with 25 μg/ml digitonin in isolation buffer (150 mM NaCl and 50 mM HEPES) for 10 min on ice. Samples were then centrifuged at $800 \times g$ for 5 min at 4 °C. Supernatants were then further centrifuged at $23,000 \times g$ for 10 min at 4 °C and pellets were discarded. Samples were normalized for protein concentration using a BioRad protein assay (BioRad). DNA was extracted from equal amounts of cytosolic fractions using TIANamp Genomic DNA Kit (TIAN-GEN), following the manufacturer's instructions. mtDNA copy number was assessed using ddPCR (Biorad) using specific probes directed against mtND1 and mtD-loop.

## Immunofluorescence staining analysis

Macrophages were stimulation with LPS and ATP or Nig. ASC were stained with a secondary antibody conjugated to either Alexa Fluor 488 and the nuclei were stained with DAPI. The cells were examined under a confocal laser microscope (LSM980; Carl Zeiss).

## In vivo LPS challenge

C57BL/B6 mice (6 weeks old) were injected i.p. with ML324 (10 mg/kg) or dimethyl sulfoxide (DMSO). After 4 h, mice were injected i.p. with 10 mg/kg LPS or PBS. After 90 min, the mice were euthanized, and blood was collected to measure serum cytokines; IL-1β, TNF-α, and IL-6 by ELISA. Similarly, *Kdm4b*$^{+/+}$ and *Kdm4b*$^{-/-}$ mice (females, 6 weeks old) were injected i.p. with 10 mg/kg LPS or PBS. After 2 h, the mice were euthanized, and blood was collected to measure serum cytokines; IL-1β, TNF-α, and IL-6 by ELISA.

## In vivo peritonitis

C57BL/B6 mice (6 weeks old) were intraperitoneally injected with ML324 (10 mg/kg) or DMSO. After 4 h, the mice were injected i.p. with 1 mg of alum for 16 h or MSU for 6 h. *Kdm4b*$^{+/+}$ or *Kdm4b*$^{-/-}$ mice (female, 6 weeks old) were injected i.p. with 1 mg alum for 16 h. The mice were euthanized, and their peritoneal cavities were washed with 6 ml of PBS. Peritoneal exudate cells (PECs) were analyzed using a fluorescence-activated cell sorting (FACS) instrument (Cytoflex; Beckman Coulter, Brea, CA, USA).

## Folic acid (FA)-induced acute tubular necrosis (ATN)

C57BL/B6 mice (female, 6 weeks old) were injected i.p. with ML324 (10 mg/kg) or DMSO. After 4 h, the mice were injected i.p. with 250 mg/kg FA. After 36 h, the mice were euthanized, and their kidneys were dissected. The kidneys were partially fixed in 4% paraformaldehyde, embedded in paraffin, sectioned, stained with hematoxylin and eosin (H&E) and periodic acid-Schiff (PAS), and examined using light microscopy for histological changes. Renal inflammation was scored (based on the proportion of renal parenchyma by PAS staining). The scoring was as follows: not obvious (score 0), less than 10% (score 1), 10–25% (score 2), 25–50% (score 3), or >50% (score 4). Partial tissue samples were ground and disrupted for immunoblotting.

## Viral pathogenesis

C57BL/B6 mice (females, 8 weeks old) were injected i.p. with ML324 (10 mg/kg) or DMSO. After 4 h, mice were injected i.p. with VSV-GFP for 18 h. The mice were euthanized, and blood was

### The paper explained

**Problem**

NLRP3 inflammasome, as the archetypical molecular driver of inflammatory response, plays crucial roles in host defense and the maintenance of cellular homeostasis. The demethylation of histone 3 lysine 9 trimethylation (H3K9me3, the repressive mark for euchromatic genes) is critical for activating gene transcription. However, whether H3K9 demethylation is required for the induction of proinflammatory cytokines remains largely unknown.

**Results**

We discover that histone demethylase lysine-specific demethylase 4B (KDM4B) mediates the demethylation of H3K9me3 at the *Nlrp3* promoter to induce NLRP3 expression. Concordantly, both *Kdm4b* deficiency and the selective KDM4 inhibitor ML324 inhibit NLRP3 inflammasome activation and ameliorate NLRP3-dependent inflammatory diseases in vivo. Moreover, high glucose upregulates KDM4B to promote NLRP3 inflammasome activation and IL-1β secretion, and therefore aggravates viral infection-induced aberrant inflammation.

**Impact**

Our work identifies KDM4B as a novel and selective therapeutic target for treating NLRP3-dependent inflammatory diseases. By demonstrating that ML324, a KDM4B inhibitor, can dampen harmful inflammation, we pave the way for developing novel targeted therapies. This strategy could be particularly beneficial for patients with conditions like diabetes, in which controlling glucose-triggered inflammation is critical.

collected to measure serum cytokines: IL-1β, TNF-α, and IL-6 by ELISA. The lungs of the mice were dissected, partially fixed in 4% paraformaldehyde, embedded in paraffin, sectioned, stained with H&E, and examined by light microscopy for histological changes. For the pathological assessment of viral-induced lung injury, the modified scoring system for viral-induced lung injury was utilized, which encompasses five critical evaluation dimensions: viral-related pathological characteristics of the lung parenchyma, alveolar structural destruction, the extent of inflammatory cell infiltration, pulmonary hemorrhage and edema, and the severity of pulmonary fibrosis

## Statistical analysis

All data are presented as the mean ± SEM of three independent experiments. Statistical significance was determined using an unpaired two-tailed Student's $t$ test, with a $P$ value < 0.05 was considered statistically significant. All statistical analyses were performed using Prism 8.

# Data availability

All data supporting the findings of this study are available within the article. This study includes no data deposited in external repositories.

The source data of this paper are collected in the following database record: biostudies:S-SCDT-10_1038-S44321-026-00373-0.

# Peer review information

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

## Acknowledgements

This work was supported by grants from the National Natural Science Foundation of China (82125020, 82201947, and 82301988), Shandong Provincial Natural Science Foundation (ZR2023ZD57 and ZR2022QH032), and Young Talent of Lifting Engineering for Science and Technology in Shandong, China (SDAST2025QTA081). We thank the Translational Medicine Core Facility of Shandong University for consultation and instrument availability that supported this work.

## Author contributions

**Li Tong**: Data curation; Formal analysis; Writing—original draft. **Hui Song**: Data curation; Writing—original draft. **Yuan Gao**: Data curation. **Danhui Qin**: Data curation. **Caiwei Wang**: Data curation; Formal analysis. **Qi Li**: Data curation. **Yue Fu**: Software; Formal analysis. **Chunyuan Zhao**: Formal analysis. **Zhendong Ying**: Formal analysis. **Dailing Chen**: Formal analysis. **Chengjiang Gao**: Supervision. **Chaofeng Han**: Supervision. **Wei Zhao**: Conceptualization; Supervision; Writing—original draft; Writing—review and editing. **Ying Qin**: Formal analysis; Supervision; Writing—original draft; Writing—review and editing. **Lei Zhang**: Resources; Supervision.

Source data underlying figure panels in this paper may have individual authorship assigned. Where available, figure panel/source data authorship is listed in the following database record: biostudies:S-SCDT-10_1038-S44321-026-00373-0.

## Disclosure and competing interests statement

The authors declare no competing interests.

# Expanded View Figures

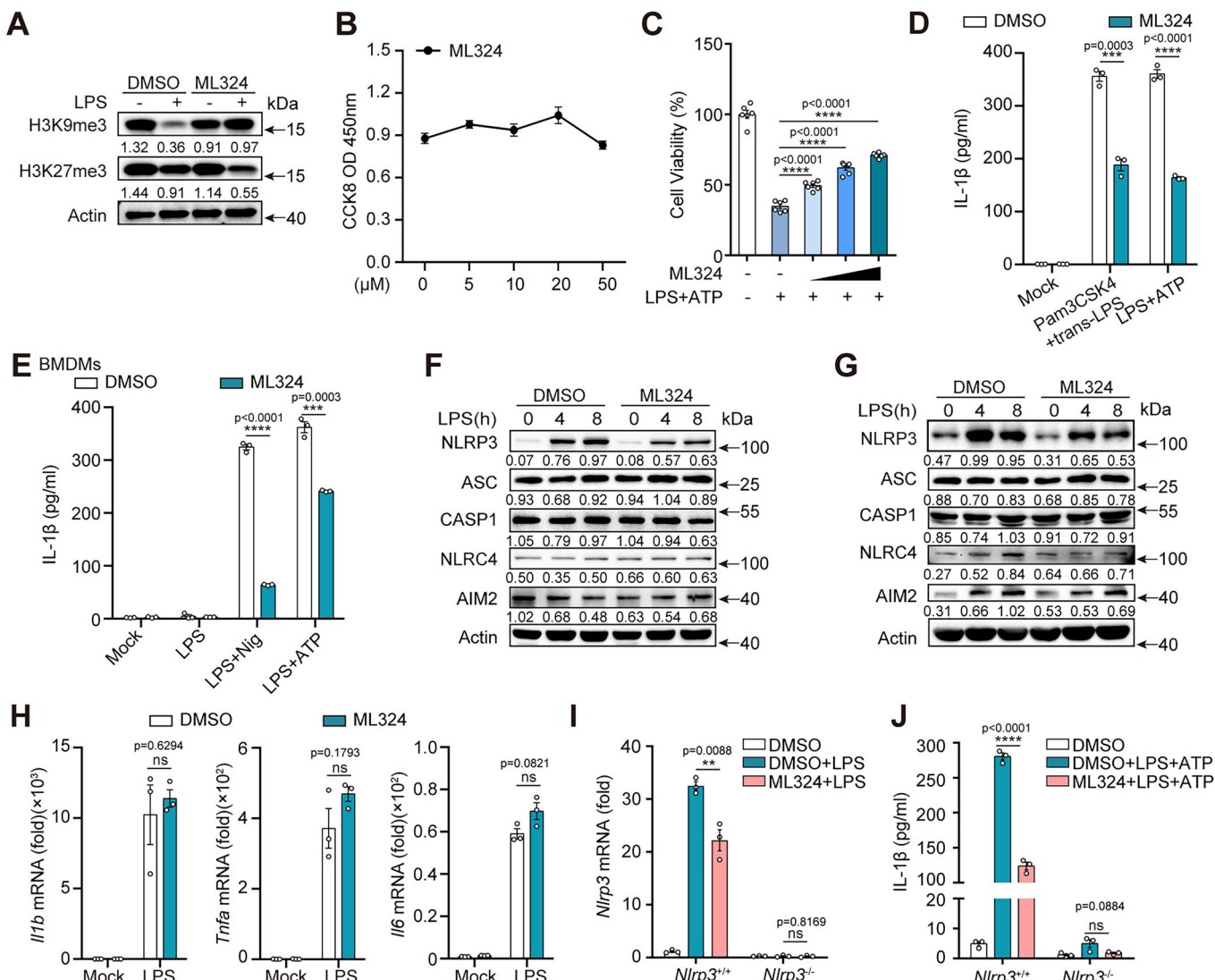

**Figure EV1.  ML324 inhibits the expression of NLRP3.**

(**A**) Western blot analysis of H3K9me3 and H3K27me3 in mouse PMs treated with DMSO or ML324 (20 μM) for 4 h, followed by LPS stimulation for 2 h. Sizes in kDa are indicated on the right. (**B, C**) Cell Counting Kit-8 (CCK8) analysis of cell viability of mouse PMs treated with DMSO or increasing amounts of ML324 (5, 10, 20, and 50 μM) for 12 h (**B**), the mouse PMs treated with DMSO or increasing amounts of ML324 (5, 10, and 20 μM) for 4 h, followed by stimulation with LPS for 6 h and subsequent stimulation with ATP for 40 min (**C**). $n = 3$ samples per group. (**D, E**) ELISA analysis of IL-1β secretion in mouse PMs (**D**) or BMDMs (**E**) treated with DMSO or ML324 (20 μM) for 4 h, stimulation with Pam3CSK4 for 6 h, followed by transfected with LPS for 2 h, or stimulation with LPS for 6 h and subsequent stimulation with ATP or Nig for 40 min. $n = 3$ samples per group. (**F, G**) Western blot analysis of NLRP3, ASC, CASP1, NLRC4, and AIM2 in mouse PMs (**F**) or human THP-1 cells (**G**) treated with DMSO or ML324 (20 μM) for 4 h, followed by stimulation with LPS for the indicated time periods. Sizes in kDa are indicated on the right. (**H**) RT-PCR analysis of *Il1b*, *Il6* and *Tnfa* mRNA expression in mouse PMs treated with DMSO or ML324 for 4 h, followed by stimulation with LPS for 2 h, $n = 3$ samples per group. (**I**) RT-PCR analysis of *Nlrp3* mRNA expression in *Nlrp3*[+/+] or *Nlrp3*[−/−] mouse PMs followed by LPS stimulation for 2 h, $n = 3$ samples per group. (**J**) ELISA analysis of IL-1β secretion in *Nlrp3*[+/+] or *Nlrp3*[−/−] mouse PMs, followed by priming with LPS for 6 h and subsequent stimulation with ATP for 40 min, $n = 3$ samples per group. Results were obtained from three independent experiments. Data are shown as mean ± SEM in (**B–E, H–J**). ns (not significant), $P > 0.05$; **$P < 0.01$; ***$P < 0.001$; ****$P < 0.0001$ (two-tailed unpaired *t* test). Source data are available online for this figure.

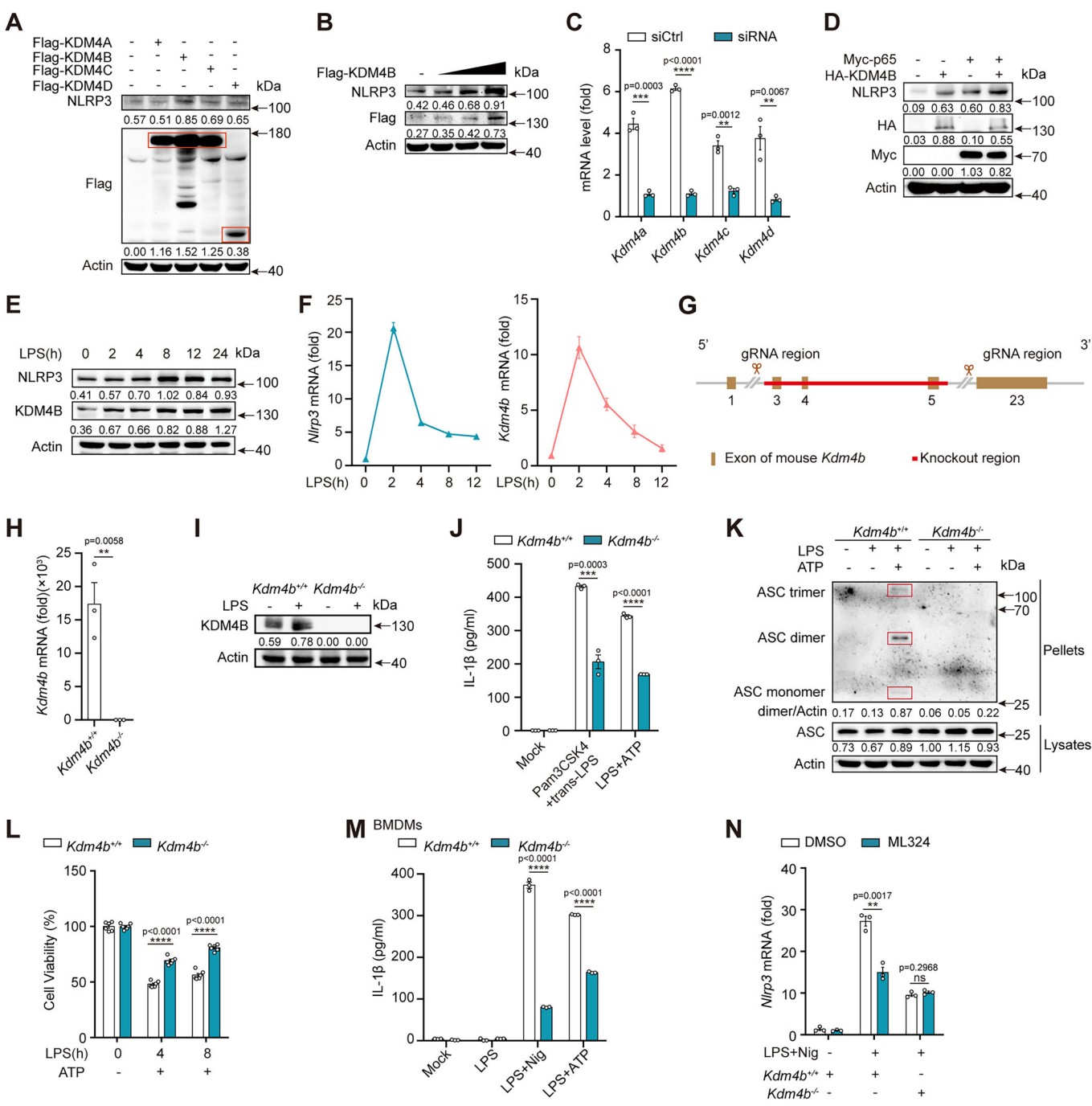

**Figure EV2.  KDM4B promotes NLRP3 expression.**

(**A**) Western blot analysis of NLRP3 expression in extracts from HEK293T cells transfected with Flag-KDM4A, KDM4B, KDM4C, and KDM4D. The red rectangles highlight the expression of target proteins. Sizes in kDa are indicated on the right. (**B**) Western blot analysis of NLRP3 expression in extracts from HEK293T cells transfected with an increasing amounts of Flag-KDM4B (0, 0.5, 1, and 2 μg). Sizes in kDa are indicated on the right. (**C**) RT-PCR analysis of *Kdm4a/b/c/d* expression in Ctrl siRNA or *Kdm4a/b/c/d* siRNA-transfected mouse PMs, $n = 3$ samples per group. (**D**) Western blot analysis of NLRP3 in HEK293T cells transfected with the indicated plasmids. Sizes in kDa are indicated on the right. (**E, F**) Western blot (**E**) or RT-PCR (**F**) analysis of NLRP3 and KDM4B expression in LPS-treated mouse PMs for various times, $n = 3$ samples per group. (**G**) Structural diagram of the knockout region in *Kdm4b$^{-/-}$* mice, provided by Cyagen Biosciences Inc. (Guangzhou, China). (**H, I**) RT-PCR (**H**, $n = 3$ samples per group) or Western blot (**I**) analysis of KDM4B in *Kdm4b$^{+/+}$* or *Kdm4b$^{-/-}$* mouse PMs. (**J**) ELISA analysis of IL-1β secretion in *Kdm4b$^{+/+}$* or *Kdm4b$^{-/-}$* mouse PMs, stimulation with Pam3CSK4 for 6 h, followed by transfected with LPS for 2 h. $n = 3$ samples per group. (**K**) Western blot analysis of ASC oligomerization in *Kdm4b$^{+/+}$* or *Kdm4b$^{-/-}$* mouse PMs, followed by priming with LPS for 6 h and subsequent stimulation with ATP for 40 min. The red rectangles highlight the different oligomeric states of ASC. Sizes in kDa are indicated on the right. (**L**) Cell Counting Kit-8 (CCK8) analysis of cell viability of *Kdm4b$^{+/+}$* or *Kdm4b$^{-/-}$* mouse PMs treated with LPS for the indicated time periods and subsequent stimulation with ATP for 40 min. $n = 6$ samples per group. (**M**) ELISA analysis of IL-1β secretion in *Kdm4b$^{+/+}$* or *Kdm4b$^{-/-}$* mouse BMDMs, stimulation with LPS for 6 h and subsequent stimulation with ATP or Nig for 40 min. $n = 3$ samples per group. (**N**) RT-PCR analysis *Nlrp3* in *Kdm4b$^{+/+}$* or *Kdm4b$^{-/-}$* mouse PMs pretreated with DMSO or ML324 (20 μM) for 4 h, followed by priming with LPS for 6 h and subsequent stimulation with Nig for 40 min. $n = 3$ samples per group. Results were obtained from three independent experiments. Data is presented as mean ± SEM in (**C, F, H, J, L–N**). ns (not significant), $P > 0.05$; **$P < 0.01$; ***$P < 0.001$; ****$P < 0.0001$ (two-tailed unpaired $t$ test). Source data are available online for this figure.

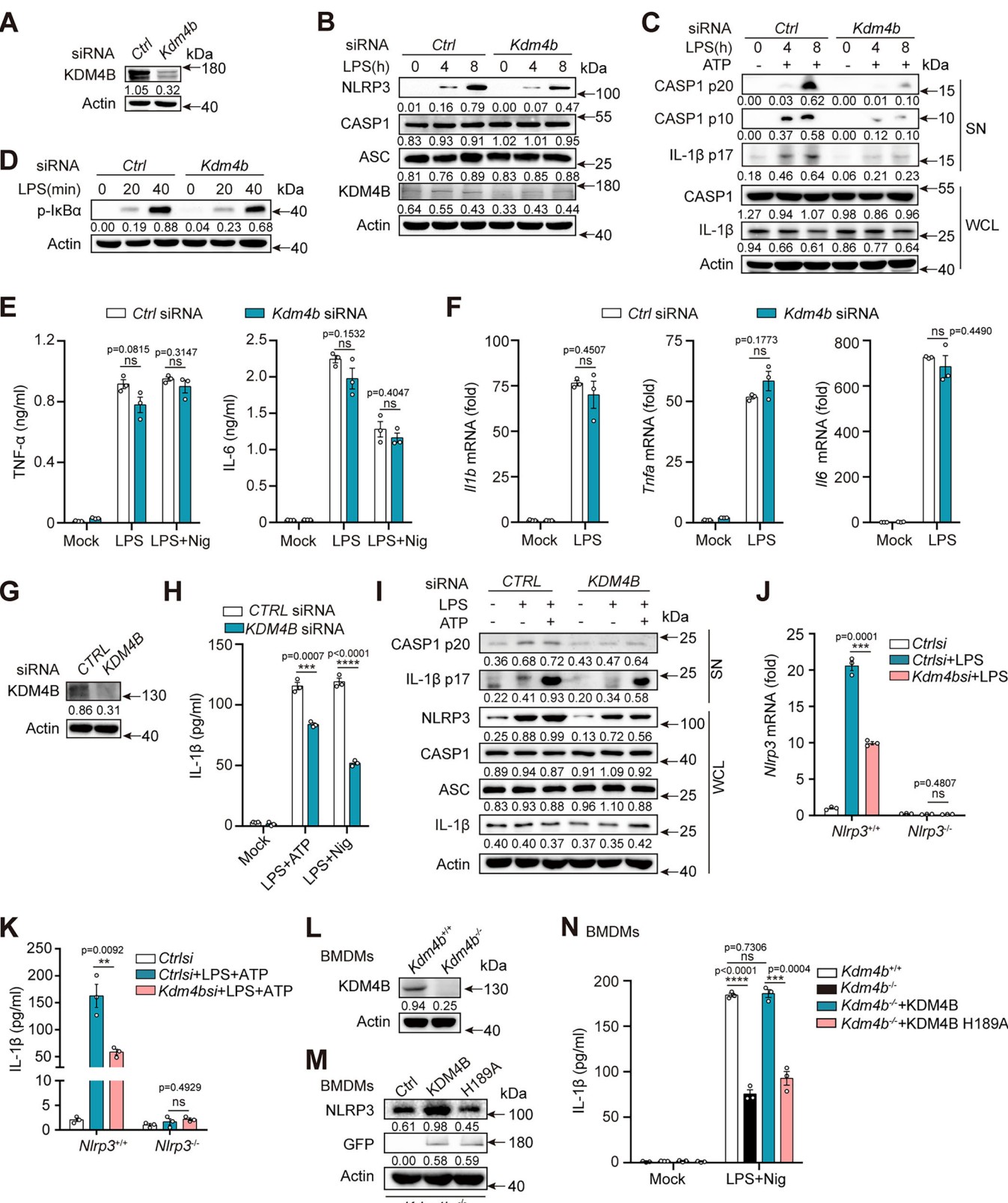

◀ **Figure EV3.** *Kdm4b* **knockdown restrains NLRP3 inflammasome activation.**

(A) Western blot analysis of KDM4B in Ctrl siRNA- or *Kdm4b* siRNA-transfected mouse PMs. Sizes in kDa are indicated on the right. (B) Western blot analysis of NLRP3, CASP1, ASC, and KDM4B in Ctrl siRNA- or *Kdm4b* siRNA-transfected mouse PMs stimulated with LPS for the indicated time periods. Sizes in kDa are indicated on the right. (C) Western blot analysis of CASP1 cleavage in Ctrl siRNA- or *Kdm4b* siRNA-transfected mouse PMs, followed by priming with LPS for 6 h and subsequent stimulation with ATP for 40 min. Sizes in kDa are indicated on the right. (D)Western blot analysis of p-IκBα in Ctrl siRNA- or *Kdm4b* siRNA-transfected mouse PMs stimulated with LPS for the indicated time periods. Sizes in kDa are indicated on the right. (E) ELISA analysis of TNF-α, and IL-6 secretion in Ctrl siRNA- or *Kdm4b* siRNA-transfected mouse PMs followed by priming with LPS for 6 h and then stimulated with Nig for 40 min, $n = 3$ samples per group. (F) RT-PCR analysis of *Il1b*, *Tnfa*, and *Il6* mRNA expression in Ctrl siRNA- or *Kdm4b* siRNA-transfected mouse PMs followed by stimulation with LPS for 2 h, $n = 3$ samples per group. (G) Western blot analysis of KDM4B in CTRL siRNA- or *KDM4B* siRNA-transfected THP-1 cells. Sizes in kDa are indicated on the right. (H) ELISA analysis of IL-1β secretion in CTRL siRNA- or *KDM4B* siRNA-transfected THP-1 cells followed by priming with LPS for 6 h and then stimulated with Nig or ATP for 40 min, $n = 3$ samples per group. (I) Western blot analysis of CASP1 cleavage p20, NLRP3, CASP1 and ASC in CTRL siRNA or KDM4B siRNA-transfected THP-1 cells, followed by priming with LPS for 6 h and subsequent stimulation with ATP for 40 min. Sizes in kDa are indicated on the right. (J) RT-PCR analysis of *Nlrp3* mRNA expression in Ctrl siRNA- or *Kdm4b* siRNA-transfected mouse PMs from *Nlrp3*$^{+/+}$ or *Nlrp3*$^{-/-}$ mice, followed by LPS stimulation for 2 h, $n = 3$ samples per group. (K) ELISA analysis of IL-1β secretion in Ctrl siRNA (Ctrlsi)- or *Kdm4b* siRNA (*Kdm4b*si)-transfected mouse PMs from *Nlrp3*$^{+/+}$ or *Nlrp3*$^{-/-}$ mice, followed by priming with LPS for 6 h and subsequent stimulation with ATP for 40 min (L + A), $n = 3$ samples per group. (L) Western blot analysis of KDM4B and NLRP3 in *Kdm4b*$^{+/+}$ or *Kdm4b*$^{-/-}$ BMDMs. Sizes in kDa are indicated on the right. (M) Western blot analysis of KDM4B and GFP in Ctrl, KDM4B-GFP, or KDM4B-H189A-GFP-transfected *Kdm4b*$^{-/-}$ BMDMs. Sizes in kDa are indicated on the right. (N) ELISA analysis of IL-1β secretion in Ctrl, KDM4B-GFP, or KDM4B-H189A-GFP-transfected *Kdm4b*$^{+/+}$ or *Kdm4b*$^{-/-}$ BMDMs, followed by priming with LPS for 6 h and subsequent stimulation with Nig for 40 min, $n = 3$ samples per group. Sizes in kDa are indicated on the right. Results were obtained from three independent experiments. Data are shown as mean ± SEM in (**E, F, H, J, K, N**), ns (not significant), $P > 0.05$; **$P < 0.01$; ***$P < 0.001$; ****$P < 0.0001$ (two-tailed unpaired *t* test). Source data are available online for this figure.

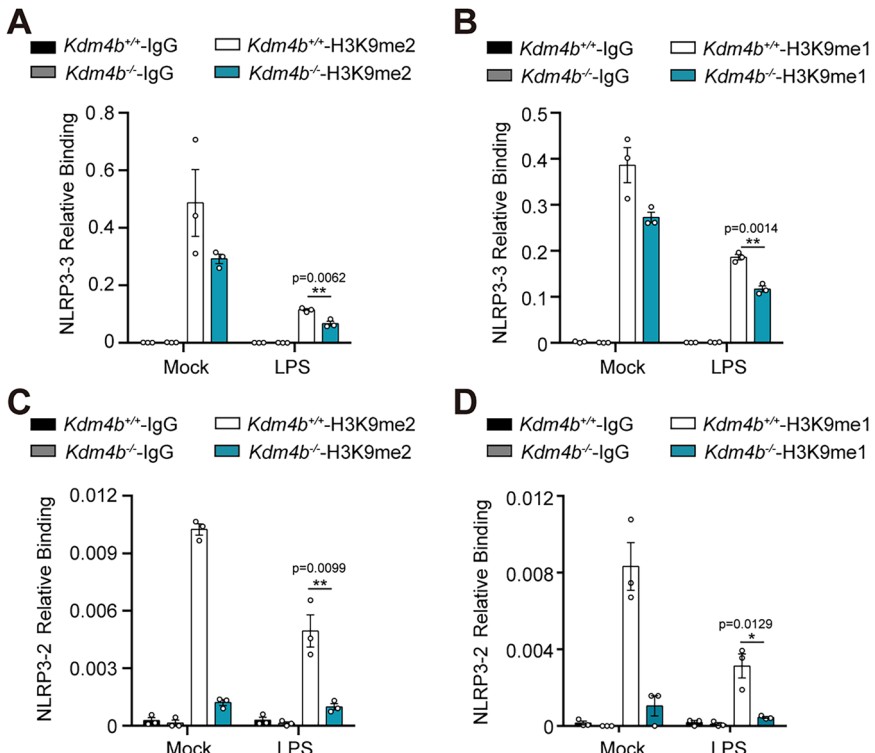

**Figure EV4. _Kdm4b_ deficiency inhibits H3K9me2 and H3K9me1 at NLRP3-2 and NLRP3-3 promoter regions.**

(**A–D**) _Kdm4b_$^{+/+}$ or _Kdm4b_$^{-/-}$ mouse PMs were stimulated with LPS, and then prepared for the ChIP-qPCR assay performed using antibodies to H3K9me2 Ab (**A**, **C**) or H3K9me1 Ab (**B**, **D**) at the corresponding region of the _Nlrp3_ promoter. Data are presented as mean ± SEM. *P < 0.05, **P < 0.01 (two-tailed unpaired _t_ test), n = 3 samples per group. Results were obtained from three independent experiments. Source data are available online for this figure.

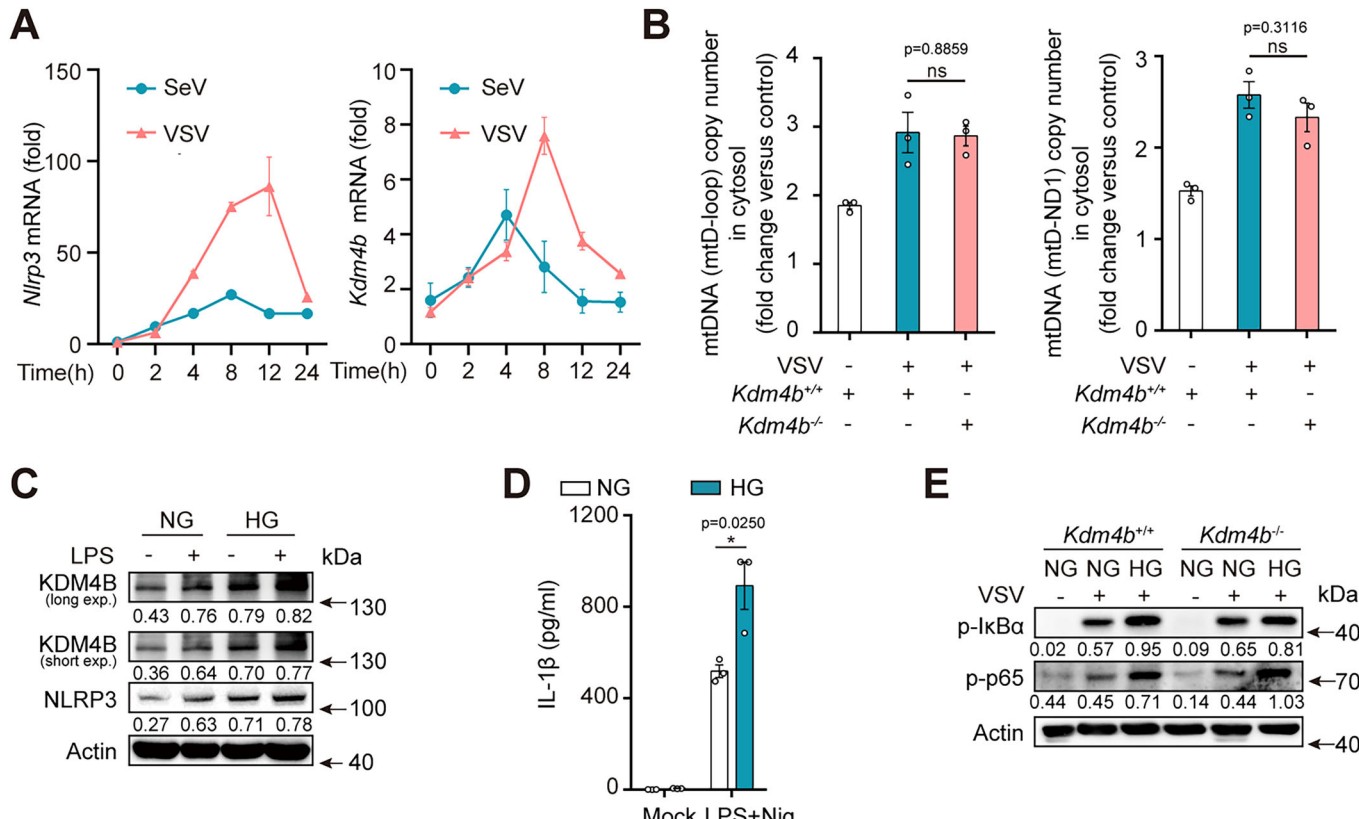

**Figure EV5. NLRP3 inflammasome activation was upregulated by high glucose.**

(A) RT-PCR analysis of NLRP3 and KDM4B expression in mouse PMs treated with SeV or VSV for various times, $n = 3$ samples per group. (B) Quantification of mtDNA copy number by ddPCR using mtD-loop or mtND1 probe, from isolated cytosolic fractions of $Kdm4b^{+/+}$ or $Kdm4b^{-/-}$ mouse PMs, $n = 3$ samples per group. (C) Western blot analysis of KDM4B and NLRP3 in mouse PMs in glucose-free medium treated with NG or HG, followed with LPS for 6 h. Sizes in kDa are indicated on the right. (D) ELISA analysis of IL-1β secretion in mouse PMs in glucose-free medium treated with NG or HG, followed by priming with LPS for 6 h and subsequent stimulation with Nig for 40 min, $n = 3$ samples per group. (E) Western blot analysis of p-IκBα and p-p65 of $Kdm4b^{+/+}$ or $Kdm4b^{-/-}$ mouse PMs in glucose-free medium and then stimulated with NG or HG, followed by VSV infection. Sizes in kDa are indicated on the right. Results were obtained from three independent experiments. Data are presented as mean ± SEM in (A, B, D). ns (not significant), $P > 0.05$; *$P < 0.05$ (two-tailed unpaired $t$ test). Source data are available online for this figure.

