## [Peer Review File · EMBO Molecular Medicine]

Histone demethylase KDM4B epigenetically controls NLRP3 expression to enhance inflammatory responses

Li Tong, Hui Song, Yuan Gao, Danhui Qin, Caiwei Wang, Qi Li, Yue Fu, Chunyuan Zhao, Zhendong Ying, Dailing Chen, Chengjiang Gao, Chaofeng Han, Wei Zhao, Ying Qin, and Lei Zhang

Corresponding author(s): Wei Zhao (wzhao@sdu.edu.cn), Ying Qin (yingqin@sdu.edu.cn), Lei Zhang (1818@sdhospital.com.cn)

Review Timeline:

Submission Date:	2nd Jun 25
Editorial Decision:	1st Jul 25
Revision Received:	31st Oct 25
Editorial Decision:	26th Nov 25
Revision Received:	15th Dec 25
Accepted:	23rd Dec 25

Editor: Zeljko Durdevic

Transaction Report:

1st Jul 2025

Dear Prof. Zhao,

Thank you for the submission of your manuscript to EMBO Molecular Medicine. We have now received feedback from the three reviewers who agreed to evaluate your manuscript. As you will see from the reports, all three referees recognize potential interest of the study, but they also raise serious concerns that should be addressed in a major revision. If you would like to discuss further the points raised by the referees, I am available to do so via email or video. Let me know if you are interested in this option.

We would welcome the submission of a revised version within three months for further consideration. Please let us know if you require longer to complete the revision.

I look forward to receiving your revised manuscript.

Yours sincerely,

Zeljko Durdevic

We require:

- 1) A .docx formatted version of the manuscript text (including legends for main figures, EV figures and tables). Please make sure that the changes are highlighted to be clearly visible.
- 2) Individual production quality figure files as .eps, .tif, .jpg (one file per figure). For guidance, download the 'Figure Guide PDF': (<https://www.embopress.org/page/journal/17574684/authorguide#figureformat>).
- 3) A .docx formatted letter INCLUDING the reviewers' reports and your detailed point-by-point responses to their comments. As part of the EMBO Press transparent editorial process, the point-by-point response is part of the Review Process File (RPF), which will be published alongside your paper.
- 4) A complete author checklist, which you can download from our author guidelines (<https://www.embopress.org/page/journal/17574684/authorguide#submissionofrevisions>). Please insert information in the checklist that is also reflected in the manuscript. The completed author checklist will also be part of the RPF.
- 5) Please note that all corresponding authors are required to supply an ORCID ID for their name upon submission of a revised manuscript.
- 6) It is mandatory to include a 'Data Availability' section after the Materials and Methods. Before submitting your revision, primary

datasets produced in this study need to be deposited in an appropriate public database, and the accession numbers and database listed under 'Data Availability'. Please remember to provide a reviewer password if the datasets are not yet public (see <https://www.embopress.org/page/journal/17574684/authorguide#dataavailability>).

12) Author contributions: You will be asked to provide CRediT (Contributor Role Taxonomy) terms in the submission system. These replace a narrative author contribution section in the manuscript.

13) A Conflict of Interest statement should be provided in the main text.

14) Every published paper now includes a 'Synopsis' to further enhance discoverability. Synopses are displayed on the journal webpage and are freely accessible to all readers. They include a short stand first (maximum of 300 characters, including space) as well as 2-5 one-sentences bullet points that summarizes the paper. Please write the bullet points to summarize the key NEW findings. They should be designed to be complementary to the abstract - i.e. not repeat the same text. We encourage inclusion of key acronyms and quantitative information (maximum of 30 words / bullet point). Please use the passive voice. Please attach

these in a separate file or send them by email, we will incorporate them accordingly.

15) Include a Reagents and Tools Table as part of the Methods section, which can be downloaded from our author guidelines (<https://www.embopress.org/page/journal/17574684/authorguide#structuredmethods>)

***** Reviewer's comments *****

Referee #1 (Comments on Novelty/Model System for Author):

NLRP3 priming is known but the precise molecular mechanism is still unclear. The paper is interesting by adding an histone demethylase in the transcriptional regulation of NLRP3.

Referee #1 (Remarks for Author):

Reviewing of "Histone demethylase KDM4B epigenetically controls NLRP3 expression to enhance inflammatory responses"
The manuscript entitled "Histone demethylase KDM4B epigenetically controls NLRP3 expression to enhance inflammatory responses" by Li Tong and colleagues explores the role of the KDM4B histone demethylase in controlling NLRP3 expression. The data presented are interesting, but important controls are missing, and some interpretations are questionable. The authors to claim that KDM4B is an activator of NLRP3 without showing it. This is confusing since the activator of NLRP3 in there experiments is Nigericin treatment. What they show is that KDM4B impairs the NLRP3 activation by Nigericin. This should be corrected.

The authors should address the specificity of their discovery by investigating the effect of KDM4B toward other inflammasomes by using other known inflammasome activators.

I.79: Use of peritoneal macrophages: extraction method drives them toward an inflammatory profile. Moreover (I.302), the authors did not detail which differentiation factor (e.g. M-CSF) was used after extraction and seeding of peritoneal cell exudate. In some experiments (I.138), BMDM were used instead of Peritoneal Macrophages, without explanation. For consistency, the authors should consider keeping one primary mouse macrophage model.

The authors should indicate clearly for each panel what is the n=? for each experiment, the stats used and if they are showing results of the replicates in a single experiment or a representative experiment of n=?.

I.83: Fig.1B: Timing between inhibitor and LPS treatments is not mentioned.

Fig 1C: The authors must measure IL_6 and TNF as control.

Fig 1D: The authors must show total IL-1b as a control in the same WB and could show the cleaved IL-b in the supernatant

I.85: Fig.1E: According to the graph, IL-6 tends to decrease with increasing doses of ML324.

I.86: May specify NLRP3 protein expression.

(I.92: Fig.1H: There is no positive control on the same graph, cannot know if the inhibitor worked well.)

I.113: "greatly" seems a bit too much, "significantly" is more appropriate.

Fig.1A, D, S2D, E, 2B, E, G, M, S4A, B, C, D, G, I, M, 4G, 6A, B, E, K, S6C, E, 7A, H, I: To support the authors' claim, the author should consider adding a relative quantification of the WB bands normalized to actin, showing them either using histograms or with numbers under each band.

I.119: Fig.S3C: Poor WB image quality. the authors could have used LPS in order to induce the protein's expression.

I.129: Fig.S4C: Casp1 p10 and p20 staining are of bad quality. The authors might consider replacing this WB.

Fig2: Title should be like KDM4B ko impairs inflammasome activation by Nigericin.

I.136: Fig.2L: Fold change of the control is not ~1.

Fig 4H and Fig5N: The size of the mice groups must be indicated in the figure legend.

Fig5N: the authors should add a group + MCC950 and ML324+MCC950 in the experiment. Alternatively, they could use NLRP3 KO mice + ML324 if available.

The authors should use the ML324 inhibitor in KDM4B ko mice. Are the data similar to WT + ML324 or the survival is increased??

I.138: Fig.S4N: Lack of statistical analysis between the "Kdm4b-/- +KDM4B H189A" and the "Kdm4b-/- +KDM4B" conditions to support the authors' claim.

I.192: Fig.S6B: Fold change of the control is not ~1.

I.193: Fig.6G: Lack of statistical analysis between the "DMSO-VSV" and the "ML324-VSV" conditions to support the authors' claim.

I.199: Fig.6M: The authors should quantify tissue damage

I.200: Fig.6N: Misinterpretation of the figure result, please correct.

I.220: Fig.7I: Not so sure about the reversal of HG effects on NLRP3 protein levels (requires quantification). The authors may

consider modifying the sentence.

I.284: About the potential use of the KDM4 inhibitor ML324 in inflammatory disorders, the authors may consider tempering their writings since in the KDM4 family, there are "four histone demethylases (KDM4A, KDM4B, KDM4C, and KDM4D) and two pseudogenes (KDM4E and KDM4F)" with potentially broad effects on gene expression.

Typos l.106, 112, 216

Referee #2 (Remarks for Author):

In this manuscript, the authors report that the histone demethylase KDM4B epigenetically controls NLRP3 inflammasome activation by selectively facilitating demethylation of H3K9me3 at the NLRP3 promoter region to increase its transcription. This conclusion results from both in vitro and in vivo analyses. The data are convincing and the conclusions raised by the authors are well supported by the results.

In the text, the authors wrote that ML324, a KDM4 demethylase inhibitor, inhibits NLRP3 inflammasome activation and that KDM4B enhances NLRP3 inflammasome activation. This is not really correct since both regulate NLRP3 expression at the transcription level but NLRP3 inflammasome activation likely occurs normally. The fact that less caspase-1 is activated and that less IL-1b is produced is not a consequence of a dysfunctional NLRP3 inflammasome, but simply because less NLRP3 is expressed.

In Kdm4b^{-/-} BMDMs, do the authors observe less ASC specks in immunofluorescence in comparison to WT BMDMs after NLRP3 inflammasome activation or are they smaller? Likewise, after cross-linking followed by WB, is there less ASC oligomerization in Kdm4b^{-/-} BMDMs?

Line 200. The authors wrote that ML-324-treated mice were more susceptible to VSV infection than the control mice (Fig. 6N). ML-324-treated mice die actually less than the control mice after infection so it means, according to me, that they are less susceptible to VSV infection. In the case of VSV infection, I believe that the virus triggers the death of the animals, not the inflammation unlike in Fig. 4B. Since in the presence of ML324, less IL-1b is produced as shown in Fig. 6L, the viral replication is then expected not to be prevented so an increased lethality should be observed, as well as an increased tissue damage in the lung. Here, the opposite is observed, how do the authors explain these observations? Are similar results obtained with Kdm4b^{-/-} mice?

Other comments:

When the supernatant was analyzed by WB, I am surprised that p17 of IL-1b is not shown as well as in the whole cell lysates.

In Figure 2M, is it possible to show the expression level of KDM4B and of the mutant?

KDM4B expression seems to be regulated by NF-κB since LPS stimulation increases its expression (Fig S2E). Can the authors confirm?

Both infection with VSV and SeV increase KDM4B expression (Fig 6A and 6B) likely as a result of NF-κB activation downstream of TLR3 or RIG-I stimulation. How does the glucose regulate expression of KDM4B?

Minor commentss

Line 96. ML324, not ML423.

Referee #3 (Comments on Novelty/Model System for Author):

This manuscript presents a comprehensive investigation into the epigenetic regulation of NLRP3 inflammasome activation by histone demethylase KDM4B. The authors provide evidence that KDM4B selectively demethylates H3K9me3 at the Nlrp3 promoter, thereby enhancing its transcription and promoting inflammasome activation. Using both genetic (Kdm4b knockout mice) and pharmacological (ML324) tools, the authors demonstrate the physiological relevance of KDM4B in various models of

inflammation, including viral infection and hyperglycemia. The study addresses a timely and important question in innate immunity and epigenetic regulation and offers a potentially valuable therapeutic target.

Referee #3 (Remarks for Author):

Major Comments:

1. ML324 is a pan-KDM4 inhibitor. While the authors convincingly show that KDM4B is critical for Nlrp3 expression, additional data should be included to exclude off-target effects. In particular, did ML324 alter the expression or activity of other KDM4 isoforms? Was H3K27 methylation at the Nlrp3 or other inflammatory gene promoters affected?
2. Although AIM2 and NLRC4 inflammasome activation were assessed, the role of KDM4B/ML324 in NLRP1 and Pyrin inflammasome activation should also be evaluated to establish selective inflammasome targeting.
3. The manuscript focuses on canonical activation. The authors are encouraged to investigate whether KDM4B plays a role in non-classical NLRP3 activation (e.g., through LPS-only stimulation in human monocytes or caspase-1-mediated pathways in mice).
4. While IL-1 β and caspase-1 are assessed, the cleavage of gasdermin D (GSDMD), a key pyroptotic effector, should also be evaluated. Additionally, direct cell death assays (e.g., LDH release, PI uptake) would strengthen the evidence for pyroptosis modulation by KDM4B/ML324.
5. Since pyroptosis is one form of inflammatory cell death, the authors are encouraged to test whether ML324 impacts other cell death pathways such as ferroptosis (e.g., spermine/spermidine stimulation, see PMID: 40425585) or necroptosis (e.g., TNF + zVAD stimulation).
6. The authors show findings in murine models and THP-1 cells, but it would significantly strengthen the translational relevance to include validation using human primary monocytes.
7. In survival experiments (Figure 4H, 5B), the protective effect of ML324 is promising. Including Nlrp3-deficient mice or MCC950-treated groups as benchmarks would provide a valuable comparison to determine the therapeutic potential of ML324.

Minor Comments:

1. The schematic model currently in Supplementary Figure S7 effectively summarizes the findings and should be moved to the main figure panel.
2. The manuscript contains several minor grammatical issues and awkward phrasing. For instance, revise "abnormal of NLRP3 inflammasome activation" to "abnormal NLRP3 inflammasome activation." A thorough proofreading or professional editing would improve readability.
3. The inclusion of densitometric quantification for key immunoblots (e.g., caspase-1, NLRP3, GSDMD) would increase transparency and data robustness.

Referee #1 (Comments on Novelty/Model System for Author):

NLRP3 priming is known but the precise molecular mechanism is still unclear. The paper is interesting by adding an histone demethylase in the transcriptional regulation of NLRP3.

Referee #1 (Remarks for Author):

Reviewing of "Histone demethylase KDM4B epigenetically controls NLRP3 expression to enhance inflammatory responses"

The manuscript entitled "Histone demethylase KDM4B epigenetically controls NLRP3 expression to enhance inflammatory responses" by Li Tong and colleagues explores the role of the KDM4B histone demethylase in controlling NLRP3 expression. The data presented are interesting, but important controls are missing, and some interpretations are questionable.

Answer: We appreciate your precious time in reviewing our manuscript. We have carefully revised the manuscript and performed several additional experiments according to your valuable suggestions, and in doing so believe we have strengthened the mechanistic details and physiological relevance of the findings.

The authors to claim that KDM4B is an activator of NLRP3 without showing it. This is confusing since the activator of NLRP3 in there experiments is Nigericin treatment. What they show is that KDM4B impairs the NLRP3 activation by Nigericin. This should be corrected.

Answer: Thanks for the valuable suggestions and we changed it in the revised manuscript.

The authors should address the specificity of their discovery by investigating the effect of KDM4B toward other inflammasomes by using other known inflammasome activators.

Answer: Thanks for your comments. We have already investigated the specificity of KDM4B by examining its impact on AIM2 and NLRC4 inflammasome activation (Fig. 2I). Our results demonstrate that KDM4B selectively regulates NLRP3 inflammasome activation without affecting AIM2 or NLRC4 inflammasome activation, confirming the specificity of our discovery.

Figure 2I

l.79: Use of peritoneal macrophages: extraction method drives them toward an inflammatory profile. Moreover (l.302), the authors did not detail which differentiation factor (e.g. M-CSF) was used after extraction and seeding of peritoneal cell exudate. In some experiments (l.138), BMDM were used instead of Peritoneal Macrophages, without explanation. For consistency, the authors should consider keeping one primary mouse macrophage model.

Answer: Thanks for your comments. We appreciate the reviewer's comment regarding the use of differentiation factors. In our study, no additional differentiation factors (such as M-CSF) were added during or after the extraction and seeding of peritoneal cell exudate because peritoneal macrophages are terminally differentiated innate immune cells that maintain their functional identity in vitro without requiring further differentiation induction. We have provided detailed methodological descriptions for BMDM differentiation (using M-CSF) in the revised Methods section. In response, we have standardized our approach by using BMDMs throughout all experiments to ensure model consistency and reproducibility. *Kdm4b* deficiency inhibited NLRP3 activation-induced IL-1 β secretion in BMDMs. We added these new data in the revised manuscript (Fig. EV 1E, and 2M).

Figure EV.1E

Figure EV.2M

The authors should indicate clearly for each panel what is the n=? for each experiment, the stats used and if they are showing results of the replicates in a single experiment or a representative experiment of n=?.

Answer: Thank you for your valuable comment. We have now clearly indicated in the figure legends for each panel the sample size (n), the statistical methods used, and whether the data represent technical replicates from a single experiment or results from multiple independent experiments.

l.83: Fig.1B: Timing between inhibitor and LPS treatments is not mentioned.

Answer: Thanks for the reviewer's valuable correction. we have now clearly indicated the precise timing between the inhibitor and LPS treatments in the legend of Figure 1B.

Fig 1C: The authors must measure IL_6 and TNF as control.

Answer: Thanks for your comments. The measurements of IL-6 and TNF- α were already included as controls in our original study, and these results are presented in Fig. 1D, and 2G.

Fig 1D: The authors must show total IL-1b as a control in the same WB and could show the cleaved IL-b in the supernatant

Answer: Thank you for this constructive suggestion. We have now included the total IL-1 β as a control in the same Western blot, and the cleaved IL-1 β in the supernatant has also been added, as suggested. We added these new data in the revised manuscript (Fig. 1E).

l.85: Fig.1E: According to the graph, IL-6 tends to decrease with increasing doses of ML324.

Answer: We appreciate the reviewer's observation. The statistical analysis yielded a p-value of 0.07, which indicates that the observed variations in IL-6 levels did not reach conventional statistical significance ($p < 0.05$). This result is consistent with our description of stable IL-6 expression under the experimental conditions. As requested, we have provided the complete raw data (Response Table 1) and have also uploaded the data set to a public repository for full transparency and verification.

Response Table 1.

	P values compared to the DMSO-LPS-ATP group			
DMSO	0.191129	0.198118	0.199194	
DMSO+LPS+ATP	1.214516	1.503495	1.471774	
ML324-5 μ m	1.645161	1.624731	1.687366	0.051862114
ML324-10 μ m	1.742204	0.80672	1.307527	0.716826624
ML324-20 μ m	1.144892	1.051613	1.240054	0.077777144
ML324-50 μ m	1.134946	0.942204	1.217204	0.07151763

l.86: May specify NLRP3 protein expression.

Answer: Thanks for the valuable suggestions. We have included relative quantification of the NLRP3 protein bands normalized to actin and annotated the numerical values onto the respective figures in the revised manuscript (Fig.1A, D; EV1A, C, D; EV2B, D, E, I; 2B, E, G, M; EV3A, B, C, D, G, I, L, M; 4G; 6A, B, E, K; EV4C, E; 7A, E, H, I).

(l.92: Fig.1H: There is no positive control on the same graph, cannot know if the inhibitor worked well.)

Answer: We thank the reviewer for this suggestion. We have now included positive controls (LPS + ATP or Nig treatment) in the same graph for Fig. 1H to verify the efficacy of the inhibitor. These new data, which confirm the expected inflammasome activation under positive control conditions, have been added to the revised manuscript (Fig. 1H).

l.113: "greatly" seems a bit too much, "significantly" is more appropriate.

Answer: Thanks for the valuable suggestions and we changed it in the revised manuscript.

Fig. 1A, D, S2D, E, 2B, E, G, M, S4A, B, C, D, G, I, M, 4G, 6A, B, E, K, S6C, E, 7A, H, I: To support the authors' claim, the author should consider adding a relative quantification of the WB bands normalized to actin, showing them either using histograms or with numbers under each band.

Answer: Thanks for the valuable suggestions. We have included relative quantification of the WB bands normalized to actin and annotated the numerical values onto the respective figures in the revised manuscript (Fig. 1A, D; EV1A, C, D; EV2B, D, E, I; 2B, E, G, M; EV3A, B, C, D, G, I, L, M; 4G; 6A, B, E, K; EV4C, E; 7A, E, H, I).

l.119: Fig.S3C: Poor WB image quality. the authors could have used LPS in order to induce the protein's expression.

Answer: We thank the reviewer for the suggestion. We have now repeated the Western blot experiment for Fig. S3C with LPS stimulation to enhance the expression of KDM4B, which has significantly improved the image quality. The updated figure has been included in the revised manuscript (Fig. EV2I).

l.129: Fig.S4C: Casp1 p10 and p20 staining are of bad quality. The authors might consider replacing this WB.

Answer: Thanks for the valuable suggestions. We have repeated the experiment and obtained higher quality images for both Casp1 p10 and p20 staining. The updated, clearer Western blot images have replaced the previous ones in Fig. EV3C within the revised manuscript.

Figure EV. 3C

Fig2: Title should be like KDM4B ko impairs inflammasome activation by Nigericin.

Answer: We thank the reviewer for this suggestion. We have revised the title of Fig. 2 to "Kdm4b deficiency impairs NLRP3 inflammasome activation" to more accurately reflect that the experiments utilized multiple NLRP3 agonists (including Nigericin and ATP).

l.136: Fig.2L: Fold change of the control is not ~1.

Answer: We appreciate the reviewer's observation. Fold change of the control is ~1 in Fig.2L (Response Table 2). As requested, we have provided the complete raw data and have also uploaded the data set to a public repository for full transparency and verification.

Response Table 2.

Ctrl	1.000000	4.794731	0.621790	4.189180	0.774561	1.734576
KDM4B	7.994815	6.186730	13.378120	17.542770	13.526320	17.60325
K189	7.324836	0.263998	7.328780	1.218513	0.149337	0.307268

Fig 4H and Fig5N: The size of the mice groups must be indicated in the figure legend.

Answer: Thanks for the reviewer's valuable correction. we have clearly indicated the size of

the mice groups in the legend of Fig. 4 and Fig. 5.

Fig5N: the authors should add a group + MCC950 and ML324+MCC950 in the experiment.

Alternatively, they could use NLRP3 KO mice + ML324 if available.

Answer: We thank the reviewer for this suggestion. As suggested by the reviewer, we have supplemented the experiment with an additional group treated with ML324 + MCC950 in mice. The results demonstrated that following LPS challenge, ML324 treatment did not affect the secretion of IL-1 β , TNF- α , and IL-6 in serum of MCC950-treated mice, nor did it impact mouse survival rates. These new data have been incorporated into the revised manuscript as Fig. 4I and J.

The authors should use the ML324 inhibitor in KDM4B ko mice. Are the data similar to WT + ML324 or the survival is increased??

Answer: Thanks for the valuable suggestions. We have conducted experiments using ML324 in *Kdm4b* KO mice. The results demonstrated that ML324 treatment did not significantly affect the secretion of IL-1 β , TNF- α , and IL-6 in serum or the survival rate in *Kdm4b*-deficient mice following LPS challenge. This indicates that the anti-inflammatory and protective effects of ML324 are specifically dependent on KDM4B inhibition, as the compound loses its efficacy in the absence of the target. These data have been included as Fig. 5G, and H in the revised manuscript.

l.138: Fig.S4N: Lack of statistical analysis between the "Kdm4b^{-/-} +KDM4B H189A" and the "Kdm4b^{-/-} +KDM4B" conditions to support the authors' claim.

Answer: We appreciated very much for the suggestion. We have performed the recommended statistical analysis between the "Kdm4b^{-/-} + KDM4B H189A" and "Kdm4b^{-/-} + KDM4B" groups. The results confirm that KDM4B, but not the catalytically inactive KDM4B H189A mutant, significantly enhanced NLRP3 inflammasome activation and NLRP3 expression, supporting our original conclusion. The updated statistical details have been added to Fig. EV3M and N in the revised manuscript.

l.192: Fig.S6B: Fold change of the control is not ~1.

Answer: We thank the reviewer for this observation. The fold change of the control group in Fig. S6B is not precisely 1 because the mtDNA copy number was first normalized to the nuclear DNA content within each individual sample to account for variations in cell number or DNA extraction efficiency, and was then normalized to the total protein concentration of the original sample to further correct for mitochondrial abundance per unit of tissue or cellular material. We have provided a comprehensive description of the data analysis methods employed in our study.

l.193: Fig.6G: Lack of statistical analysis between the "DMSO-VSV" and the "ML324-VSV" conditions to support the authors' claim.

Answer: We appreciated for the valuable suggestion. We have now performed statistical analysis between the "DMSO-VSV" and "ML324-VSV" groups in Fig. 6G. The results show that ML324 treatment significantly increased H3K9me3 enrichment at the *Nlrp3* promoter region ($p < 0.05$), supporting our original conclusion. The updated statistical details have been

added to Fig. 6G in the revised manuscript.

Figure6.G

l.199: Fig.6M: The authors should quantify tissue damage

Answer: We appreciated for the valuable suggestion. We have quantified the tissue damage. The results demonstrated a significant increase in the histopathological score in the injury group compared to controls ($p < 0.01$), providing objective assessment of tissue damage severity. These data have been added to Fig. 6M in the revised manuscript.

Figure6.M

l.200: Fig.6N: Misinterpretation of the figure result, please correct.

Answer: We appreciated for the helpful suggestion and reword the following paragraph in the manuscript: However, ML324-treated mice were less susceptible to VSV infection and showed enhanced resistance compared to the WT mice (Fig. 6N).

l.220: Fig.7I: Not so sure about the reversal of HG effects on NLRP3 protein levels (requires quantification). The authors may consider modifying the sentence.

Answer: Thanks for the valuable suggestions. We have included relative quantification of the WB bands normalized to actin and annotated the numerical values onto the respective figures in the revised manuscript (Fig.7I).

l.284: About the potential use of the KDM4 inhibitor ML324 in inflammatory disorders, the authors may consider tempering their writings since in the KDM4 family, there are "four histone demethylases (KDM4A, KDM4B, KDM4C, and KDM4D) and two pseudogenes (KDM4E and KDM4F)" with potentially broad effects on gene expression.

Answer: We appreciated for the valuable suggestion. In response, we have conducted additional experiments administering ML324 to *Kdm4b* knockout mice. The results demonstrated that ML324 treatment did affect the serum levels of IL-1 β , TNF- α , and IL-6, nor did it alter survival rates following LPS challenge in *Kdm4b*-deficient mice (Fig. 5G and H). This indicates that the anti-inflammatory and protective effects of ML324 are specifically dependent on KDM4B inhibition, as the compound loses its efficacy in the absence of the target rather than off-target effects on other KDM4 family members. These data have been included as Fig. 5G and H in the revised manuscript.

Typos l.106, 112, 216

Answer: We sincerely thank the reviewer for their helpful suggestion. We have thoroughly reviewed lines 106, 112, and 216 in our manuscript and made appropriate corrections to ensure accuracy.

Referee #2 (Remarks for Author):

In this manuscript, the authors report that the histone demethylase KDM4B epigenetically controls NLRP3 inflammasome activation by selectively facilitating demethylation of H3K9me3 at the NLRP3 promoter region to increase its transcription. This conclusion results from both in vitro and in vivo analyses. The data are convincing and the conclusions raised by the authors are well supported by the results.

Answer: We appreciate your precious time in reviewing our manuscript. We have carefully revised the manuscript and performed several additional experiments according to your valuable suggestions, and in doing so believe we have strengthened the mechanistic details and physiological relevance of the findings.

In the text, the authors wrote that ML324, a KDM4 demethylase inhibitor, inhibits NLRP3 inflammasome activation and that KDM4B enhances NLRP3 inflammasome activation. This is not really correct since both regulate NLRP3 expression at the transcription level but NLRP3 inflammasome activation likely occurs normally. The fact that less caspase-1 is activated and that less IL-1 β is produced is not a consequence of a dysfunctional NLRP3 inflammasome, but simply because less NLRP3 is expressed.

Answer: We thank the reviewer for this insightful comment. We agree that our original description could have been more precise. As suggested, we have revised the manuscript to clarify that ML324 inhibits NLRP3 expression at the transcriptional level, and that KDM4B enhances NLRP3 expression. This clarification has been made throughout the revised manuscript (line 258-260).

In Kdm4b^{-/-} BMDMs, do the authors observe less ASC specks in immunofluorescence in comparison to WT BMDMs after NLRP3 inflammasome activation or are they smaller? Likewise, after cross-linking followed by WB, is there less ASC oligomerization in Kdm4b^{-/-} BMDMs?

Answer: Thanks for the suggestions. We have performed both immunofluorescence and Western blot analysis. *Kdm4b* deficiency impaired ASC speck formation, and ASC oligomerization (Fig. 2F, and EV2K). These data have been included as Fig. 2F, and EV2K in

the revised manuscript.

*Line 200. The authors wrote that ML-324-treated mice were more susceptible to VSV infection than the control mice (Fig. 6N). ML-324-treated mice die actually less than the control mice after infection so it means, according to me, that they are less susceptible to VSV infection. In the case of VSV infection, I believe that the virus triggers the death of the animals, not the inflammation unlike in Fig. 4B. Since in the presence of ML324, less Il-1b is produced as shown in Fig. 6L, the viral replication is then expected not to be prevented so an increased lethality should be observed, as well as an increased tissue damage in the lung. Here, the opposite is observed, how do the authors explain these observations? Are similar results obtained with *Kdm4b*^{-/-} mice?*

Answer: We appreciated for the helpful suggestion. The statement has been corrected to: "However, ML324-treated mice were less susceptible to VSV infection and showed enhanced resistance compared to the WT mice (Fig. 6N)." This aligns with the survival data. Furthermore, as suggested, we found that *Kdm4b*^{-/-} mice also exhibited significantly higher survival rates after VSV infection compared to WT mice (Fig. 6O). During the early stages of viral infection, the host may develop a cytokine storm, a dysregulated and excessive immune response characterized by the rapid release of high levels of pro-inflammatory cytokines and chemokines, which can lead to multi-organ dysfunction syndrome (MODS) and death (Sefik et al, 2022; Teijaro et al, 2011). KDM4B inhibition or genetic deficiency attenuates virus-induced hyperinflammation and cytokine production, reducing immunopathological damage and thereby improving survival. This aligns with models where immunomodulation, rather than enhanced viral clearance, can be beneficial in severe infections characterized by excessive inflammation. These data have been included as Fig. 6O in the revised manuscript.

Figure6.O

Other comments:

When the supernatant was analyzed by WB, I am surprised that p17 of IL-1 β is not shown as well as in the whole cell lysates.

Answer: We appreciated very much for the valuable suggestions. We have detected both IL-1 β expression (in whole cell lysates, WCL) and its secretion (in supernatants, SN) in our experiments. These data have been included as Fig. 1E, 2E, 6E, 6K, EV3C, and EV3I in the revised manuscript.

In Figure 2M, is it possible to show the expression level of KDM4B and of the mutant?

Answer: We appreciated for the valuable suggestion. We have included Western blot analysis the expression levels of KDM4B and the mutant in the Fig. 2N. These data have been included as Fig. 2N in the revised manuscript.

KDM4B expression seems to be regulated by NF- κ B since LPS stimulation increases its expression (Fig S2E). Can the authors confirm?

Both infection with VSV and SeV increase KDM4B expression (Fig 6A and 6B) likely as a result of NF- κ B activation downstream of TLR3 or RIG-I stimulation. How does the glucose regulate expression of KDM4B?

Answer: Thanks for your great questions. Previous studies have established that KDM4B expression is subject to transcriptional regulation by multiple transcription factors, including ER α , HIF-1 α , p53, and NF- κ B (p65) (Yang et al, 2010; Beyer et al, 2008; Castellin et al, 2017; Yi et al, 2021). Therefore, LPS stimulation, viral infection, and high glucose levels may upregulate KDM4B expression through activation of the aforementioned transcriptional regulators. This area remains open for further investigation. We discussed this issue in the manuscript.

Minor commentss

Line 96. ML324, not ML423.

Answer: Thanks for the valuable suggestions and we changed it in the revised manuscript.

Referee #3 (Comments on Novelty/Model System for Author):

This manuscript presents a comprehensive investigation into the epigenetic regulation of NLRP3 inflammasome activation by histone demethylase KDM4B. The authors provide evidence that KDM4B selectively demethylates H3K9me3 at the Nlrp3 promoter, thereby enhancing its transcription and promoting inflammasome activation. Using both genetic (Kdm4b knockout mice) and pharmacological (ML324) tools, the authors demonstrate the physiological relevance of KDM4B in various models of inflammation, including viral infection and hyperglycemia. The study addresses a timely and important question in innate immunity and epigenetic regulation and offers a potentially valuable therapeutic target.

Answer: We appreciate your precious time in reviewing our manuscript. We have carefully revised the manuscript and performed several additional experiments according to your valuable suggestions, and in doing so believe we have strengthened the mechanistic details and physiological relevance of the findings. Please find below our responses to the comments and suggestions.

Referee #3 (Remarks for Author):

Major Comments:

1. ML324 is a pan-KDM4 inhibitor. While the authors convincingly show that KDM4B is critical for Nlrp3 expression, additional data should be included to exclude off-target effects. In particular, did ML324 alter the expression or activity of other KDM4 isoforms? Was H3K27 methylation at the Nlrp3 or other inflammatory gene promoters affected?

Answer: We thank the reviewer for raising this important point. Our data demonstrate that ML324 lost its effects on inhibiting IL-1 β secretion and *Nlrp3* expression in *Kdm4b*^{-/-} PMs and mice (Fig. 2K, Fig. 5G, 5H, and EV2J), indicating that its anti-inflammatory actions are specifically mediated through KDM4B inhibition rather than off-target effects on other KDM4 isoforms. Furthermore, additional ChIP-seq analysis confirmed that ML324 treatment did not alter H3K27 methylation at the *Nlrp3* promoter or other inflammatory gene promoters (Response Fig. 1). These results collectively support the specificity of ML324 for targeting KDM4B.

Response Figure 1

Response Fig. 1: ChIP-qPCR analysis for H3K27me3 at the corresponding region of the *Nlrp3* promoter in mouse PMs were treated with DMSO or ML324 (20 μ M) for 4 h, followed by stimulation with LPS for 2 h, n=3 samples per group.

2. Although AIM2 and NLRC4 inflammasome activation were assessed, the role of KDM4B/ML324 in NLRP1 and Pyrin inflammasome activation should also be evaluated to establish selective inflammasome targeting.

Answer: We appreciate the reviewer's suggestion. While we attempted to assess this experimentally, specific activators for NLRP1 inflammasome (such as anthrax lethal toxin) and Pyrin inflammasome (such as *Clostridium difficile* toxin B) are strictly regulated and unavailable due to current biosafety policies. Alternatively, we examined whether KDM4B inhibition affects the transcriptional levels of NLRP1 and Pyrin. Our results show that ML324 treatment and *Kdm4b* deficiency did not alter the mRNA expression of NLRP1(*Nod*) or Pyrin (*Mefv*) in PMs (Response Fig. 2), suggesting that KDM4B selectively modulates NLRP3 inflammasome activation at the transcriptional level (via H3K9me3 demethylation) rather than broadly targeting all inflammasome sensors. These findings further support the specificity of KDM4B-dependent regulation toward NLRP3, as reported in our study.

Response Figure 2

Response Fig. 2: RT-PCR analysis of *Nod1* and *Mefv* mRNA expression in *Kdm4b*^{+/+} or *Kdm4b*^{-/-} mouse PMs or the mice PMs treated with DMSO or ML324 (20μM), followed by stimulated with LPS. n=3 samples per group.

3. The manuscript focuses on canonical activation. The authors are encouraged to investigate whether KDM4B plays a role in non-classical NLRP3 activation (e.g., through LPS-only stimulation in human monocytes or caspase-11-mediated pathways in mice).

Answer: We appreciated for the valuable suggestion. We have now investigated the role of KDM4B in non-canonical NLRP3 inflammasome activation. Our results demonstrated that ML324 treatment and *Kdm4b* deficiency significantly attenuated NLRP3 inflammasome activation induced by LPS transfection (Fig. EV1D, and EV2J). These findings confirm that KDM4B broadly regulates NLRP3-driven inflammation beyond the canonical pathway, likely through epigenetic modulation of NLRP3. These data have been included as Fig. EV1D, and EV2J in the revised manuscript.

4. While *IL-1 β* and *caspase-1* are assessed, the cleavage of gasdermin D (GSDMD), a key pyroptotic effector, should also be evaluated. Additionally, direct cell death assays (e.g., LDH release, PI uptake) would strengthen the evidence for pyroptosis modulation by *KDM4B/ML324*.

Answer: We appreciated for the valuable suggestion. As appropriately suggested by the reviewer, we have now evaluated GSDMD cleavage by Western blot and performed direct cell death assays. Our results demonstrated that both ML324 treatment and *KDM4B* deficiency significantly reduced GSDMD cleavage and decreased cell death upon NLRP3 inflammasome activation. The updated results are included in the revised manuscript (Fig. 1E, 2E, EV1C, and EV2L).

5. Since pyroptosis is one form of inflammatory cell death, the authors are encouraged to test whether ML324 impacts other cell death pathways such as ferroptosis (e.g., spermine/spermidine stimulation, see PMID: 40425585) or necroptosis (e.g., TNF + zVAD stimulation).

Answer: We appreciated for the suggestion. While we agree that exploring these pathways would be interesting, our study specifically focuses on the epigenetic regulation of inflammation (such as *KDM4B*-mediated H3K9me3 demethylation at the *Nlrp3* promoter)

rather than direct modulation of cell death execution mechanisms. The primary goal of this work was to elucidate how KDM4B inhibition suppresses NLRP3 inflammasome activation and subsequent pyroptosis by transcriptional control, rather than broadly targeting cell death pathways. However, we fully acknowledge the importance of these pathways in inflammatory contexts and will consider them for future studies.

6. The authors show findings in murine models and THP-1 cells, but it would significantly strengthen the translational relevance to include validation using human primary monocytes.

Answer: We appreciated very much for the suggestion. However, due to significant technical challenges associated with maintaining the phenotype and function of human primary monocytes in ex vivo culture, along with current limitations in accessing sufficient and ethically sourced human primary cells under our resource constraints, we are unable to incorporate these experiments in the present revision. We hope the compelling data from our murine models and THP-1 human cell line, which consistently demonstrate a specific role for KDM4B in regulating NLRP3 inflammasome activation, suffice to support our conclusions at this stage.

7. In survival experiments (Figure 4H, 5B), the protective effect of ML324 is promising. Including Nlrp3-deficient mice or MCC950-treated groups as benchmarks would provide a valuable comparison to determine the therapeutic potential of ML324.

Answer: We thank the reviewer for this suggestion. As suggested by the reviewer, we have supplemented the experiment with an additional group treated with ML324 + MCC950 in mice. The results demonstrated that following LPS challenge, ML324 treatment did not affect the secretion of IL-1 β , TNF- α , and IL-6 in serum of MCC950-treated mice, nor did it impact mouse survival rates. These new data have been incorporated into the revised manuscript as Fig. 4I, and J.

Minor Comments:

1. The schematic model currently in Supplementary Figure S7 effectively summarizes the findings and should be moved to the main figure panel.

Answer: Thanks for the valuable correction. We have moved the schematic model from Supplementary Figure S7 to the main manuscript, and it is now presented as Fig. 8.

2. The manuscript contains several minor grammatical issues and awkward phrasing. For instance, revise "abnormal of NLRP3 inflammasome activation" to "abnormal NLRP3 inflammasome activation." A thorough proofreading or professional editing would improve readability.

Answer: Thanks for the valuable suggestions. We have carefully proofread the manuscript and revised awkward phrases, including changing "abnormal of NLRP3 inflammasome activation" to "abnormal NLRP3 inflammasome activation" as suggested. The manuscript has been professionally edited to improve overall readability

3. The inclusion of densitometric quantification for key immunoblots (e.g., caspase-1, NLRP3, GSDMD) would increase transparency and data robustness.

Answer: Thanks for the valuable suggestions. We have included densitometric quantification for the key immunoblots (caspase-1, NLRP3, and GSDMD) and annotated the numerical values onto the respective figures in the revised manuscript (Fig.1A, E; EV1A, F, G; EV2A, B, D, E, I, K; 2B, E, H, N; EV3A, B, C, D, G, I, L, M; 4G; 6A, B, E, K; EV5C, E; 7A, E, H, I) .

REFERENCES

- Beyer S, Kristensen M M, Jensen K S, Johansen J V, Staller P (2008) The histone demethylases JMJD1A and JMJD2B are transcriptional targets of hypoxia-inducible factor HIF. *J Biol Chem* 283(52), 36542 – 36552.
- Castellini L, Moon E J, Razorenova O V, Krieg A J, von Eyben R, Giaccia A J (2017) KDM4B/JMJD2B is a p53 target gene that modulates the amplitude of p53 response after DNA damage. *Nucleic Acids Res* 45(7), 3674 – 3692.
- Sefik E, Qu R, Junqueira C, Kaffe E, Mirza H, Zhao J, Brewer J R, Han A, Steach H R, Israelow B, Blackburn H N et al (2022) Inflammasome activation in infected macrophages drives COVID-19 pathology. *Nature* 606(7914): 585 – 593.
- Teijaro J R, Walsh K B, Cahalan S, Fremgen D M, Roberts E, Scott F, Martinborough E, Peach R, Oldstone M B, Rosen H (2011) Endothelial cells are central orchestrators of cytokine amplification during influenza virus infection. *Cell* 146(6), 980 – 991.
- Yang J, Jubb A M, Pike L, Buffa F M, Turley H, Baban D, Leek R, Gatter K C, Ragoussis J, Harris A L (2010) The histone demethylase JMJD2B is regulated by estrogen receptor alpha and hypoxia, and is a key mediator of estrogen induced growth. *Cancer Res* 70(16), 6456 – 6466.
- Yi S J, Jang Y J, Kim H J, Lee K, Lee H, Kim Y, Kim J, Hwang S Y, Song J S, Okada H, Park J I, Kang K, Kim K (2021) The KDM4B-CCAR1-MED1 axis is a critical regulator of osteoclast differentiation and bone homeostasis. *Bone Res* 9(1), 27.

26th Nov 2025

Dear Prof. Zhao,

Thank you for the submission of your revised manuscript to EMBO Molecular Medicine. I am pleased to inform you that we will be able to accept your manuscript pending the following final amendments:

- 1) Please address the referee #1 minor point.
- 2) In the main manuscript file, please do the following:
 - Please address all comments suggested by our data editors listed below:
 - o Figure legends:
 1. Please note that the exact p values are not provided in the legends of figures 1c,f,h,i,j; 2a,c,d,j,k,l,m; 3a-c,e,f; 4a-c,f,h,i,j; 5a-d,f-h; 6c,d,f-j,l,m,o; 7b,c,d,f,g,j,k; EV-1 c,d,e,l,j; EV-2c,h,j,l,m,n; EV-3h,j,k,n; EV-4a-d; EV-5d.
 2. Please note that the error bars are not defined in the legends of figures 3a; EV-1b,j; EV-2f.
 3. Please note that the white arrows are not defined in the legend of figure 2f. This needs to be rectified.
 4. Please note that the red rectangles are not defined in the legend of figures EV-2a,k. This needs to be rectified.
 - Add up to 5 keywords.
 - Rename "Materials and Methods" to "Methods".
 - In Methods, provide the antibody dilutions that were used for each antibody.
 - Please include structured Methods section that includes a Reagents and Tools Table (should be uploaded as a separate file) followed by a Methods and Protocols section. More information on how to adhere to this format as well as downloadable templates (.docx) for the Reagents and Tools Table can be found in our author guidelines: <https://www.embopress.org/page/journal/17574684/authorguide#structuredmethods>
 - An example of a paper with Structured Methods can be found here: <https://www.embopress.org/doi/full/10.1038/s44320-024-00037-6#sec-4>
 - Please rename "Competing interests" to "Disclosure and competing interests statement". We updated our journal's competing interests policy in January 2022 and request authors to consider both actual and perceived competing interests. Please review the policy <https://www.embopress.org/competing-interests> and update your competing interests if necessary.
 - Author contributions: Please remove it from the manuscript and specify author contributions in our submission system. CRediT has replaced the traditional author contributions section because it offers a systematic machine-readable author contributions format that allows for more effective research assessment. You are encouraged to use the free text boxes beneath each contributing author's name to add specific details on the author's contribution. More information is available in our guide to authors: <https://www.embopress.org/page/journal/17574684/authorguide#authorshippinguidelines>
 - Indicate in legends number and nature of replicates and exact p= values, not a range, along with the statistical test used. To keep the figures "clear" some authors found providing an Appendix table Sx with all exact p-values preferable. You are welcome to do this if you want to.
 - In data availability statement please replace the current sentence with "This study includes no data deposited in external repositories."
 - Please align the references to the left.
 - Please rename the expanded view figures to "Figure EV1" etc.
- 3) Tables: Please remove the EV Tables from the manuscript text and upload them as two separate files. Please rename them to Table EV1 and EV2.
- 4) The Paper Explained: Please provide "The Paper Explained" and add it to the main manuscript text. Please check "Author Guidelines" for more information. <https://www.embopress.org/page/journal/17574684/authorguide#researcharticleguide>
- 5) Synopsis: Every published paper now includes a 'Synopsis' to further enhance discoverability. Synopses are displayed on the journal webpage and are freely accessible to all readers. They include separate synopsis image and synopsis text.
 - Synopsis image: Please provide a visual abstract as a high-resolution jpeg file 550 px-wide x 300-600 pixels high to illustrate your article.
 - Synopsis text: Please provide a short standfirst (maximum of 300 characters, including space) as well as 2-5 one sentence bullet points that summarise the paper as a .doc file. Please write the bullet points to summarise the key NEW findings. They should be designed to be complementary to the abstract - i.e. not repeat the same text. We encourage inclusion of key acronyms and quantitative information (maximum of 30 words / bullet point). Please use the passive voice.
 - Please check your synopsis text and image before submission with your revised manuscript. Please be aware that in the proof stage minor corrections only are allowed (e.g., typos).
- 6) As part of the EMBO Publications transparent editorial process initiative (see our Editorial at <http://embomolmed.embopress.org/content/2/9/329>), EMBO Molecular Medicine will publish online a Review Process File (RPF) to accompany accepted manuscripts. This file will be published in conjunction with your paper and will include the anonymous referee reports, your point-by-point response and all pertinent correspondence relating to the manuscript. Let us know whether you agree with the publication of the RPF and as here, if you want to remove or not any figures from it prior to publication. Please note that the Authors checklist will be published at the end of the RPF.
- 7) Please provide a point-by-point letter INCLUDING my comments as well as the reviewer's reports and your detailed

responses (as Word file).

I look forward to reading a new revised version of your manuscript as soon as possible.

Yours sincerely,

Zeljko Durdevic

Zeljko Durdevic
Senior Editor
EMBO Molecular Medicine

*** Instructions to submit your revised manuscript ***

When preparing your revised manuscript, please refer to our guidelines: <https://link.springer.com/journal/44321/submission-guidelines#cms-Revised-submissions>. We perform an initial quality control of all revised manuscripts before re-review; failure to include requested items will delay the evaluation of your revision.

We require:

- 1) A .docx formatted version of the manuscript text (including legends for main figures, EV figures and tables). Please make sure that the changes are highlighted to be clearly visible.
- 2) Individual production quality figure files as .eps, .tif, .jpg (one file per figure). For guidance, download the 'Figure Guide PDF': <https://media.springernature.com/original/springer-cms/rest/v1/content/27825798/data/v1>.
- 3) A .docx formatted letter INCLUDING the reviewers' reports and your detailed point-by-point responses to their comments. As part of the EMBO Press transparent editorial process, the point-by-point response is part of the Review Process File (RPF), which will be published alongside your paper.
- 4) A complete author checklist, which you can download from our author guidelines. Please insert information in the checklist that is also reflected in the manuscript. The completed author checklist will also be part of the RPF.
- 5) Please note that all corresponding authors are required to supply an ORCID ID for their name upon submission of a revised manuscript.
- 6) It is mandatory to include a 'Data Availability' section after the Materials and Methods. Before submitting your revision, primary datasets produced in this study need to be deposited in an appropriate public database, and the accession numbers and database listed under 'Data Availability'. Please remember to provide a reviewer password if the datasets are not yet public.

- 7) For data quantification: please specify the name of the statistical test used to generate error bars and P values, the number

(n) of independent experiments (specify technical or biological replicates) underlying each data point and the test used to calculate p-values in each figure legend. The figure legends should contain a basic description of n, P and the test applied. Graphs must include a description of the bars and the error bars (s.d., s.e.m.).

9) Our journal encourages inclusion of *data citations in the reference list* to directly cite datasets that were re-used and obtained from public databases. Data citations in the article text are distinct from normal bibliographical citations and should directly link to the database records from which the data can be accessed. In the main text, data citations are formatted as follows: "Data ref: Smith et al, 2001" or "Data ref: NCBI Sequence Read Archive PRJNA342805, 2017". In the Reference list, data citations must be labeled with "[DATASET]". A data reference must provide the database name, accession number/identifiers and a resolvable link to the landing page from which the data can be accessed at the end of the reference.

12) Author contributions: You will be asked to provide CRediT (Contributor Role Taxonomy) terms in the submission system. These replace a narrative author contribution section in the manuscript.

13) A Disclosure and competing interests statement should be provided in the main text.

14) Every published paper includes a 'Synopsis' to further enhance discoverability. Synopses are displayed on the journal webpage and are freely accessible to all readers. They include a short stand first (maximum of 300 characters, including space) as well as 2-5 one-sentences bullet points that summarizes the paper. Please write the bullet points to summarize the key NEW findings. They should be designed to be complementary to the abstract - i.e. not repeat the same text. We encourage inclusion of key acronyms and quantitative information (maximum of 30 words / bullet point). Please use the passive voice. Please attach these in a separate file or send them by email, we will incorporate them accordingly.

15) Include a Reagents and Tools Table as part of the Methods section, which can be downloaded from our author guidelines.

Each figure should be given in a separate file and should have the following resolution:
Graphs 800-1,200 DPI

Photos 400-800 DPI
Colour (only CMYK) 300-400 DPI"

*Additional important information regarding figures and illustrations can be found at
<https://media.springernature.com/original/springer-cms/rest/v1/content/27825798/data/v1>

***** Reviewer's comments *****

Referee #1 (Comments on Novelty/Model System for Author):

The manuscript has been greatly improved and has reached the standard for publication.

Referee #1 (Remarks for Author):

The authors have greatly improved the manuscript.

Minor point: It is still not clear in the text how NLRC4 activation by flagellin was performed in Fig. 1H and 2I. Please clearly indicate this point in the figure legend or in the Methods section.

Referee #2 (Remarks for Author):

The referees' comments have been addressed in a satisfactory manner. I therefore recommend acceptance for publication.

Referee #3 (Comments on Novelty/Model System for Author):

The revised manuscript has been improved and is suitable for publication.

Referee #3 (Remarks for Author):

The authors have satisfactorily addressed most of my comments.

Referee #1 (Comments on Novelty/Model System for Author):

The manuscript has been greatly improved and has reached the standard for publication.

Referee #1 (Remarks for Author):

The authors have greatly improved the manuscript.

Answer: We appreciate very much for your work in reviewing our manuscript. The insightful suggestions and comments help us to greatly improve the manuscript. According to your suggestions and comments, we carefully revised the manuscript.

Minor point: It is still not clear in the text how NLRC4 activation by flagellin was performed in Fig. 1H and 2I. Please clearly indicate this point in the figure legend or in the Methods section.

Answer: We appreciated very much for your valuable suggestions and comments. In our study, cytosolic delivery of flagellin to activate NLRC4 inflammasome in PMs was achieved using Lipofectamine 2000. Specifically, purified flagellin was complexed with Lipofectamine 2000 in serum-free Opti-MEM medium and then added to the LPS-primed PMs. This method facilitates the efficient translocation of flagellin into the cytosol, enabling its recognition by the cytosolic receptor NAIP5 and subsequent NLRC4 inflammasome assembly. The figure legends have been updated accordingly to briefly mention "transfected flagellin" in the revised manuscript.

Referee #2 (Remarks for Author):

The referees' comments have been addressed in a satisfactory manner. I therefore recommend acceptance for publication.

Answer: We appreciate very much for your work in reviewing our manuscript. The insightful suggestions and comments help us to greatly improve the manuscript.

Referee #3 (Comments on Novelty/Model System for Author):

The revised manuscript has been improved and is suitable for publication.

Referee #3 (Remarks for Author):

The authors have satisfactorily addressed most of my comments.

Answer: We appreciate very much for your work in reviewing our manuscript. The insightful suggestions and comments help us to greatly improve the manuscript.

Thank you for the submission of your revised manuscript to EMBO Molecular Medicine. I am pleased to inform you that we will be able to accept your manuscript pending the following final amendments:

Answer: We appreciated very much for your work in editing our manuscript and the valuable suggestions. We have carefully revised the manuscript, point-by-point, in accordance with the editorial feedback, and have made comprehensive revisions to address all concerns raised.

1) Please address the referee #1 minor point.

Answer: We sincerely thank Referee #1 for their positive feedback and the valuable minor point raised. We have carefully addressed this comment in the revised manuscript. Specifically, as suggested, we have made corresponding revisions in the figure legends to briefly mention "transfected flagellin". We believe this clarification has improved the clarity of our manuscript and hope it now fully addresses the referee's point.

2) In the main manuscript file, please do the following:

- Please address all comments suggested by our data editors listed below:

o Figure legends:

1. Please note that the exact p values are not provided in the legends of figures 1c,f,h,i,j; 2a,c,d,j,k,l,m; 3a-c,e,f; 4a-c,f,h,i,j; 5a-d,f-h; 6c,d,f-j,l,m,o; 7b,c,d,f,g,j,k; EV-1 c,d,e,l,j; EV-2c,h,j,l,m,n; EV-3h,j,k,n; EV-4a-d; EV-5d.

Answer: Thank you for your valuable comment. We have added the precise p-values directly on all statistically analyzed figures in the revised manuscript.

2. Please note that the error bars are not defined in the legends of figures 3a; EV-1b,j; EV-2f.

Answer: Thank you for your valuable comment. We have addressed this issue by adding the legends of figures in the revised manuscript.

3. Please note that the white arrows are not defined in the legend of figure 2f. This needs to be rectified.

Answer: Thank you for your valuable comment. We have supplemented the figure

legend to explicitly state that the white arrows indicate ASC specks.

4. Please note that the red rectangles are not defined in the legend of figures EV-2a,k. This needs to be rectified.

Answer: Thank you for pointing out the missing definition of red rectangles. We have supplemented the legends to specify that the red rectangles represent the expression locations of the target proteins.

- Add up to 5 keywords.

Answer: We thank the editor for this suggestion. We have now added the following five keywords to the manuscript: NLRP3; KDM4B; H3K9me3; Epigenetic modification; NLRP3 inflammasome.

- Rename "Materials and Methods" to "Methods".

Answer: Thanks for the valuable suggestions and we changed it in the revised manuscript.

- In Methods, provide the antibody dilutions that were used for each antibody.

- Please include structured Methods section that includes a Reagents and Tools Table (should be uploaded as a separate file) followed by a Methods and Protocols section.

More information on how to adhere to this format as well as downloadable templates (.docx) for the Reagents and Tools Table can be found in our author guidelines:

<https://www.embopress.org/page/journal/17574684/authorguide#structuredmethods>

An example of a paper with Structured Methods can be found here:

<https://www.embopress.org/doi/full/10.1038/s44320-024-00037-6#sec-4>

Answer: Thank you for your valuable guidance regarding the manuscript revisions. We have carefully addressed all the requirements.

- Please rename "Competing interests" to "Disclosure and competing interests statement". We updated our journal's competing interests policy in January 2022 and request authors to consider both actual and perceived competing interests. Please review the policy <https://www.embopress.org/competing-interests> and update your

competing interests if necessary.

Answer: Thanks for the valuable suggestions and we changed it in the revised manuscript.

- Author contributions: Please remove it from the manuscript and specify author contributions in our submission system. CRediT has replaced the traditional author contributions section because it offers a systematic machine-readable author contributions format that allows for more effective research assessment. You are encouraged to use the free text boxes beneath each contributing author's name to add specific details on the author's contribution. More information is available in our guide to authors:

<https://www.embopress.org/page/journal/17574684/authorguide#authorshipguideline>

Answer: Thanks for the valuable suggestions and we changed it in the revised manuscript.

- Indicate in legends number and nature of replicates and exact p= values, not a range, along with the statistical test used. To keep the figures "clear" some authors found providing an Appendix table Sx with all exact p-values preferable. You are welcome to do this if you want to.

Answer: We thank you for this important suggestion regarding statistical reporting. We have carefully reviewed all figure legends to ensure they now explicitly state both the number and the nature of the biological replicates, which are three independent experiments. As suggested, all exact p-values have been directly included on the figures themselves for clarity and immediate reference. Additionally, the specific statistical test used for each analysis (e.g., unpaired two-tailed Student's t-test, log rank test [Mantel-Cox]) is now clearly indicated in the respective figure legends.

- In data availability statement please replace the current sentence with "This study includes no data deposited in external repositories."

- Please align the references to the left.

- Please rename the expanded view figures to "Figure EV1" etc.

3) Tables: Please remove the EV Tables from the manuscript text and upload them as

two separate files. Please rename them to Table EV1 and EV2.

Answer: Thank you for your valuable guidance regarding the manuscript revisions. These issues have been addressed in the revised manuscript.

4) The Paper Explained: Please provide "The Paper Explained" and add it to the main manuscript text. Please check "Author Guidelines" for more information. <https://www.embopress.org/page/journal/17574684/authorguide#researcharticleguide>

Answer: Thanks for the valuable suggestions. We have now included the "The Paper Explained" section in the main manuscript text, as required by the journal's guidelines.

5) Synopsis: Every published paper now includes a 'Synopsis' to further enhance discoverability. Synopses are displayed on the journal webpage and are freely accessible to all readers. They include separate synopsis image and synopsis text.

- Synopsis image: Please provide a visual abstract as a high-resolution jpeg file 550 px-wide x 300-600 pixels high to illustrate your article.

- Synopsis text: Please provide a short standfirst (maximum of 300 characters, including space) as well as 2-5 one sentence bullet points that summarise the paper as a .doc file. Please write the bullet points to summarise the key NEW findings. They should be designed to be complementary to the abstract - i.e. not repeat the same text. We encourage inclusion of key acronyms and quantitative information (maximum of 30 words / bullet point). Please use the passive voice.

Answer: We thank the editor for this reminder. We have submitted both the Synopsis image and the Synopsis text, as required by the journal's guidelines.

6) As part of the EMBO Publications transparent editorial process initiative (see our Editorial at <http://embomolmed.embopress.org/content/2/9/329>), EMBO Molecular Medicine will publish online a Review Process File (RPF) to accompany accepted

manuscripts. This file will be published in conjunction with your paper and will include the anonymous referee reports, your point-by-point response and all pertinent correspondence relating to the manuscript. Let us know whether you agree with the publication of the RPF and as here, if you want to remove or not any figures from it prior to publication. Please note that the Authors checklist will be published at the end of the RPF.

Answer: We confirm that we agree with the publication of the RPF as presented. We have reviewed the figures and do not wish to remove any of them prior to publication; we believe they are all essential for supporting the findings of our study.

7) Please provide a point-by-point letter INCLUDING my comments as well as the reviewer's reports and your detailed responses (as Word file).

Answer: We have prepared a detailed point-by-point response letter as a Word file, which includes all comments from the editor and reviewers along with our responses. The document has been uploaded to the submission system.

23rd Dec 2025

Dear Prof. Zhao,

We are pleased to inform you that your manuscript is accepted for publication and is now being sent to our publisher to be included in the next available issue of EMBO Molecular Medicine.

You may qualify for financial assistance for your publication charges - either via a Springer Nature fully open access agreement or an EMBO initiative. Check your eligibility: <https://link.springer.com/journal/44321/how-to-publish-with-us>

Zeljko Durdevic
Senior Editor
EMBO Molecular Medicine

>>> Please note that it is EMBO Molecular Medicine policy for the transcript of the editorial process (containing referee reports and your response letter) to be published as an online supplement to each paper. If you do NOT want this, you will need to inform the Editorial Office via email immediately. More information is available here: <https://link.springer.com/partners/embo-press/editorial-policies#Peer%20review>